# Impact of nucleic acid and methylated H3K9 binding activities of Suv39h1 on its heterochromatin assembly

**Atsuko Shirai[1†], Takayuki Kawaguchi[2,3†], Hideaki Shimojo[4], Daisuke Muramatsu[1], Mayumi Ishida-Yonetani[5], Yoshifumi Nishimura[4], Hiroshi Kimura[6], Jun-ichi Nakayama[2,3*], Yoichi Shinkai[1*]**

[1]Cellular Memory Laboratory, RIKEN, Wako, Japan; [2]Division of Chromatin Regulation, National Institute for Basic Biology, Okazaki, Japan; [3]Graduate School of Natural Sciences, Nagoya City University, Nagoya, Japan; [4]Graduate School of Medical Life Science, Yokohama City University, Yokohama, Japan; [5]Laboratory for Chromatin Dynamics, RIKEN Center for Developmental Biology, Kobe, Japan; [6]Cell Biology Unit, Institute of Innovative Research, Tokyo Institute of Technology, Yokohama, Japan

**Abstract** SUV39H is the major histone H3 lysine 9 (H3K9)-specific methyltransferase that targets pericentric regions and is crucial for assembling silent heterochromatin. SUV39H recognizes trimethylated H3K9 (H3K9me3) via its chromodomain (CD), and enriched H3K9me3 allows SUV39H to target specific chromosomal regions. However, the detailed targeting mechanisms, especially for naïve chromatin without preexisting H3K9me3, are poorly understood. Here we show that Suv39h1's CD (Suv39h1-CD) binds nucleic acids, and this binding is important for its function in heterochromatin assembly. Suv39h1-CD had higher binding affinity for RNA than DNA, and its ability to bind nucleic acids was independent of its H3K9me3 recognition. Suv39h1 bound major satellite RNAs *in vivo*, and knockdown of major satellite RNAs lowered Suv39h1 retention on pericentromere. Suv39h1 mutational studies indicated that both the nucleic acid–binding and H3K9me–binding activities of Suv39h1-CD were crucial for its pericentric heterochromatin assembly. These results suggest that chromatin-bound RNAs contribute to creating SUV39H's target specificity.

**\*For correspondence:** jnakayam@nibb.ac.jp (J-iN); yshinkai@riken.jp (YS)

[†]These authors contributed equally to this work

**Competing interests:** The authors declare that no competing interests exist.

## Introduction

Heterochromatin, a highly condensed chromatin structure, can be divided into two classes: invariably constitutive heterochromatin (CH), and developmentally controlled facultative heterochromatin. CH generally consists of repetitive DNA and is transcriptionally silent, and its higher-order chromatin structure can spread, causing the heritable inactivation of genes adjacent to the CH (*Grewal and Elgin, 2002*; *Grewal and Moazed, 2003*; *Richards and Elgin, 2002*). The establishment of CH is tightly correlated with changes in posttranslational histone-tail modifications. The trimethylation of histone H3 lysine 9 (H3K9me3), which is a hallmark of CH, creates specific binding sites for H3K9me3 reader molecules such as HP1 proteins (*Bannister et al., 2001*; *Lachner and Jenuwein, 2002*; *Lachner et al., 2001*; *Nakayama et al., 2001*), and these epigenetic states are maintained throughout replication (*Audergon et al., 2015*; *Margueron and Reinberg, 2010*; *Ragunathan et al., 2015*). CH is critical for functional chromosomal domains such as the pericentromere, and epigenetic pericentromere layers are important for accurate chromosome segregation in mitosis and for genome stability (*Allshire et al., 1995*; *Peters et al., 2001*). SUV39H (1 and 2), an H3K9me3

**eLife digest** Plants, animals and fungi store much of their DNA tightly packed with proteins in a form named heterochromatin. This arrangement helps to inactivate genes that are not needed in specific cells or at specific times, and provides a way to protect the genetic material from damage. Heterochromatin tends to form when an enzyme called a lysine methyltransferase chemically modifies some of the proteins associated with the DNA, which are known as histones. This enzyme modifies only some of the histones to get the process started, while a second protein then binds to the modified histones and causes more of the DNA to become packaged up as heterochromatin.

In 2012, researchers reported that the version of the lysine methyltransferase enzyme from yeast binds to RNA molecules via a portion known as its chromodomain. Moreover, the enzyme needed to bind to RNA to help heterochromatin to form. A similar mechanism also occurs in fruit flies, another organism that is commonly studied in the laboratory. However, it was not clear if it happened in mammals like mice and humans.

Now, Shirai, Kawaguchi et al. – who include many of the researchers involved in the 2012 study – report that the corresponding enzyme from mice can also bind to RNA molecules via its chromodomain. Further experiments showed that this activity was closely linked with the enzyme's ability to target the correct histones and efficiently form heterochromatin. The first experiments were conducted using purified enzymes in the laboratory, while follow-up experiments looked at the enzyme's activity within mouse cells.

Other studies have previously reported that mutant mice lacking the lysine methyltransferase enzyme have defective heterochromatin, tend to die young and have genetic instabilities that are associated with an increased risk of tumors and male infertility. The new findings of Shirai, Kawaguchi et al. reveal that the mechanism behind the establishment of heterochromatin has mostly likely been conserved over a billion years of evolution, which is when yeast and mammals last shared a common ancestor. By revealing more about how mammalian cells can protect their DNA, these new findings could also mark an important step toward understanding and preventing birth defects that are caused when an embryo's genetic material becomes damaged.

methyltransferase in mammals (*Rea et al., 2000*), targets CH domains and intact transposons (*Bulut-Karslioglu et al., 2014*). H3K9me3 in the pericentric regions is mainly catalyzed by SUV39H (*Peters et al., 2003*). Therefore, in *Suv39h1/2* double-knockout (*Suv39h* dn) cells, the H3K9me3 levels over the pericentromere are severely diminished and HP1 proteins do not accumulate (*Lachner et al., 2001*).

The nature of CH and its epigenetic machineries are well conserved in various species, including fission yeast and flies (*Elgin and Reuter, 2013*; *Grewal and Jia, 2007*). In fission yeast *Schizosaccharomyces pombe* (*S. pombe*), the RNA interference (RNAi)-dependent pathway is essential for pericentric heterochromatin formation and maintenance, and the process of pericentric transcription is coupled with the targeting of the SUV39H homologue Clr4 to chromatin (*Castel and Martienssen, 2013*; *Goto and Nakayama, 2012*; *Grewal and Jia, 2007*; *Moazed, 2009*). In the *Drosophila* ovary, a similar small-RNA-mediated process promotes transcriptional silencing by H3K9me3 and by HP1A, an HP1-family protein in flies (*Castel and Martienssen, 2013*). However, it is not clear whether the RNAi-dependent pathway (or another RNA-based mechanism) is involved in the process of establishing and maintaining heterochromatin in mammals, such as in creating target specificities for SUV39H and SUV39H-mediated H3K9me3 formation.

Suv39h-family proteins have two functionally distinct domains, the N-terminal chromodomain (CD) and the C-terminal SET domain. The CD functions as a binding module that targets H3K9me3 marks, and the SET domain is responsible for SUV39H's enzymatic activity. Since the N-terminal CD of Clr4 is required for the efficient spread and stable inheritance of H3K9me3 marks (*Al-Sady et al., 2013*; *Noma et al., 2004*; *Ragunathan et al., 2015*; *Zhang et al., 2008*), it has been suggested that Suv39/Clr4 uses a direct read-write mechanism to maintain the H3K9me3 marks on CH. However, this mechanism alone is not sufficient for converting H3K9me3-free, naïve chromosomal regions to silent heterochromatin.

Here we demonstrated that Suv39h1-CD can bind nucleic acids, and that this binding is crucial for Suv39h1's function. Suv39h-CD prefers to bind ssRNA rather than dsDNA, and Suv39h-CD's nucleic acid binding is independent of its H3K9me3 recognition. Mutational analyses revealed that both the nucleic acid- and H3K9me3-binding activities of Suv39h1 are important for its induction of heterochromatin assembly. We also showed that Suv39h1 bound major satellite RNA *in vivo* in a manner dependent on its nucleic acid-binding activity and that knockdown of major satellite RNAs lowered Suv39h1 retention on pericentric heterochromatin. Our data suggest that chromatin-bound RNAs contribute to the targeting and retention of Suv39h1 at specific chromosomal regions.

## Results

### Suv39h1-CD can bind nucleic acids

We recently reported that Chp1, a CD protein functioning in the RNAi pathway in fission yeast, binds nucleic acid via its CD, and that this activity is required for heterochromatin assembly (*Ishida et al., 2012*). Experiments in the same study revealed that Clr4, the fission yeast homologue of SUV39H, also binds nucleic acid via its CD, although the importance of Clr4-CD's nucleic acid binding in heterochromatin assembly remains elusive. Here, to investigate whether Suv39h1 also binds nucleic acids, we produced a GST-fusion protein containing Suv39h1-CD (residues 39–105) (*Figure 1A,B*, GST-Suv39h1-CD) and conducted electrophoretic mobility-shift assays (EMSAs) using *in vitro* transcribed, single-stranded 130-nt major satellite RNA (ssRNA) as a probe. GST-fusion proteins containing fission yeast Chp1-CD (GST-Chp1-CD) or GST alone were used as controls (*Ishida et al., 2012*). As shown in *Figure 1C*, GST-Suv39h1-CD bound ssRNA robustly and far more efficiently than did GST-Chp1-CD, although how these CDs activities contribute to the overall RNA-binding efficiency of full-length proteins remains elusive. Clr4-CD's binding of nucleic acids was detected only when K9-methylated H3 peptide was added to the EMSA (*Ishida et al., 2012*); thus, although evolutionarily conserved roles need to be considered, nature of Suv39h1-CD's binding of nucleic acid appeared to be distinct from that of Clr4-CD.

We previously showed that fission yeast Chp1-CD and Clr4-CD bind both ssRNA and double-stranded (ds) DNA (*Ishida et al., 2012*). We tested whether Suv39h1-CD could also bind DNA by an EMSA using 130 bp major satellite dsDNA. Although Suv39h1-CD was able to bind dsDNA, titration analysis revealed that Suv39h1-CD had a lower binding affinity for dsDNA than for ssRNA (*Figure 1D–G*; $K_D$ = 0.35 ± 0.02 μM for ssRNA and 0.93 ± 0.05 μM for dsDNA). Although nucleosomal DNA might contribute to Suv39h1's chromatin targeting, the present study focused on Suv39h1-CD's ssRNA-binding activity.

It has been proposed that noncoding transcripts corresponding to satellite repeats are functionally linked with CH assembly and/or kinetochore formation (*Maison et al., 2011*; *Rošić et al., 2014*). To determine the type of sequences that are targeted by Suv39h1-CD, we prepared several different ssRNAs and subjected them to an RNA EMSA (*Figure 1—figure supplement 1*). Suv39h1-CD was able to bind both 130-nt alpha satellite ssRNA and 130-nt *ACTB* ORF ssRNA with a similar affinity as for 130-nt major satellite ssRNA (*Figure 1D*, *Figure 1—figure supplement 1A–C*). Suv39h1-CD bound similarly to major satellite ssRNAs with a length of 74–189 nt (*Figure 1D*, *Figure 1—figure supplement 1D–F*). Recently, Porro et al. reported that GST-SUV39H1 preferentially binds to telomeric-repeat ((UUAGGG)$_{10}$) ssRNAs and that part of SUV39H1's N terminus (residues 1–106) is responsible for this RNA-binding activity (*Porro et al., 2014*); these findings are consistent with our results. However, we found that GST-Suv39h1-CD's binding affinity for 60-nt (UUAGGG)$_{10}$ was similar to that of 74-nt major satellite ssRNAs (*Figure 1—figure supplement 1G*).

DNA–RNA hybrid formation is reported to play a role in heterochromatin assembly in fission yeast (*Nakama et al., 2012*). In addition, transcripts of major satellite repeats are produced in both sense and antisense orientation (*Bulut-Karslioglu et al., 2014*), implying existence of chromatin-associated double-stranded RNAs (dsRNAs). Therefore, we investigated whether Suv39h1-CD has a higher binding affinity for DNA-RNA hybrid strands or dsRNA than for ssRNA. However, Suv39h1-CD bound major satellite RNA-DNA hybrid strands and dsRNA with similar affinities to that of ssRNA (*Figure 1—figure supplement 1D,H,I*), indicating that RNA-DNA hybrid molecules or dsRNA do not have a unique role in this recognition. Taken together, these results suggested that Suv39h1-CD binds RNA without a major preference for particular structures or sequences.

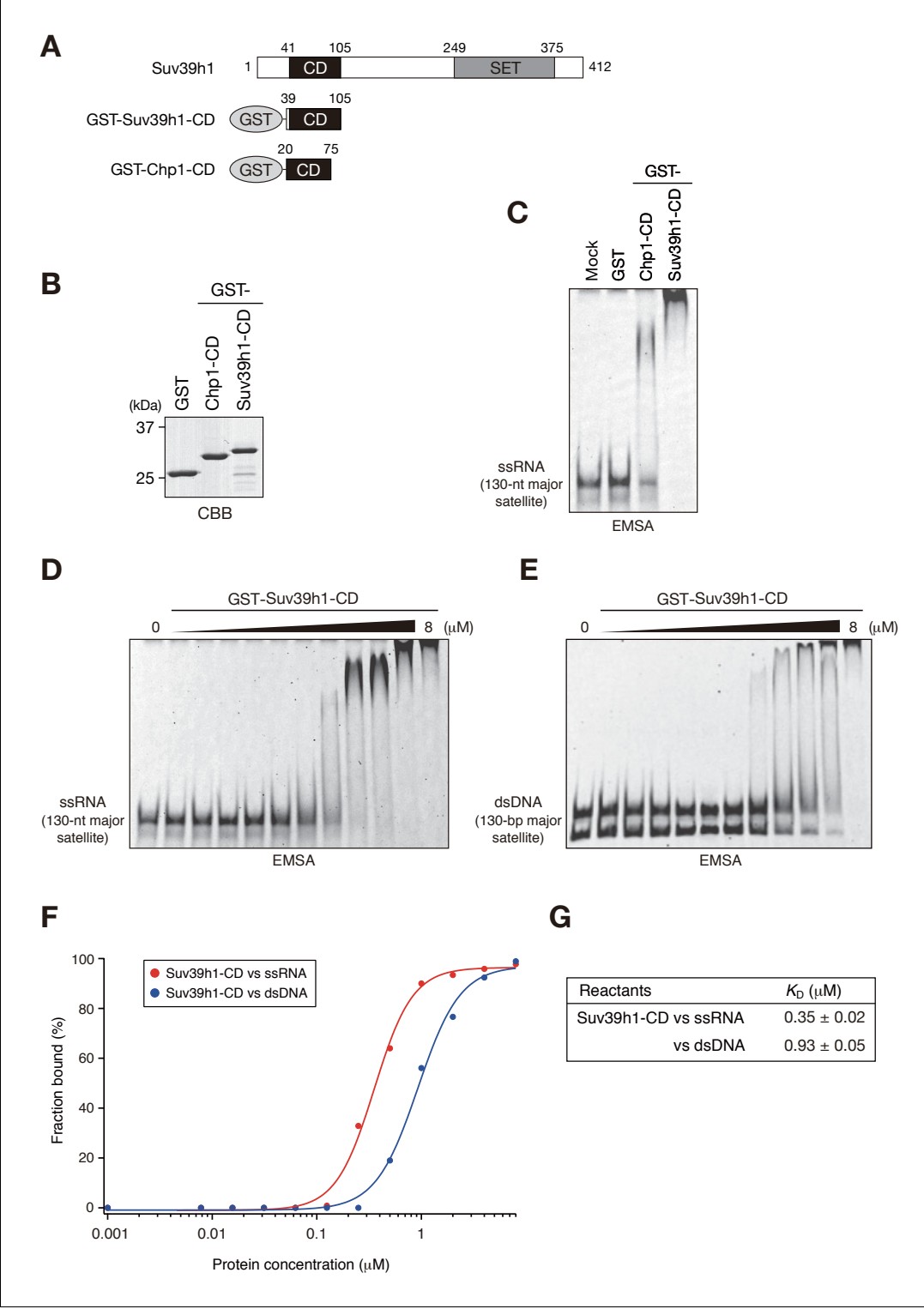

**Figure 1.** Suv39h1-CD can bind nucleic acids. (**A**) Schematic of full-length Suv39h1, GST-fused Suv39h1-CD (39–105), and *S. pombe* Chp1-CD (20–75). (**B**) The recombinant GST-fused proteins used in (**C**); these proteins were visualized by CBB staining. (**C**) An EMSA using GST-fused proteins. 8 µM of GST or GST-fusion was used in one assay. Fluorescein-labeled 130-nt major satellite ssRNA was used as a probe. (**D and E**) Titration EMSAs using GST-Suv39h1-CD (0–8 µM with 0.5-fold dilutions) incubated with (**D**) 130-nt ssRNA or (**E**) 130 bp dsDNA. DNA probe was detected as doublet bands in gels, because the number of the fluorescent dye, which was conjugated at the single or both 5'-ends, could slightly affect the migration. (**F**) The binding isotherm of Suv39h1-CD to ssRNA

*Figure 1 continued on next page*

*Figure 1 continued*

and dsDNA; plots were calculated from the unbound fractions. (**G**) The dissociation constants measured by titration ssRNA and dsDNA EMSA experiments (**D** and **E**).

The following figure supplement is available for figure 1:

**Figure supplement 1.** Characteristics of Suv39h1-full-length and -CD's RNA binding.

## Suv39h1-CD's RNA binding and methylated H3K9 recognition/binding are independent of each other

In the cases of fission yeast Chp1-CD and Clr4-CD, nucleic acid binding is coupled with H3K9me2/3 binding (*Ishida et al., 2012*). Therefore, we next examined whether Suv39h1-CD's RNA binding is linked with its H3K9me3 recognition. We first determined whether Suv39h1-CD shows a preference for methylated H3K9 peptide. As shown in *Figure 2—figure supplement 1A*, isothermal titration calorimetry (ITC) revealed that GST-Suv39h1-CD bound H3K9me3 peptide (H3K9me3) with an affinity of $K_D$ = 26 µM, but no significant binding was observed for unmodified H3 peptide (H3unmod); this was consistent with previous observations (*Wang et al., 2012*). Furthermore, a peptide pull-down assay demonstrated that GST-Suv39h1-CD had higher binding affinities for di- or trimethylated H3K9 peptide (H3K9me0 < me1 << me2 ≦ me3) (*Figure 2C*, WT).

To examine the relationship between Suv39h1-CD's RNA binding and H3K9me3 binding, we introduced an amino acid substitution at Y67; this residue, along with F43 and W64, is responsible for Suv39h1-CD's recognition of H3K9me3 (*Wang et al., 2012*) (*Figure 2A*, indicated in green). Although a Suv39h1-CD mutant with Y67A (Suv39h1-CD-Y67A, *Figure 2B*) retained residual binding activity for unmodified or K9-methylated H3 peptide, it lost its preference for di- and tri-methylated H3K9 (*Figure 2C*, Y67A). An EMSA demonstrated that Suv39h1-CD-Y67A bound ssRNA with an affinity almost the same as that of wild-type Suv39h1-CD (*Figure 2D,E*), indicating that Suv39h1-CD's RNA binding was distinct from its H3K9me3 recognition. To confirm this possibility, we added H3K9me3 peptide to the EMSA; this addition enhanced Chp1-CD's RNA binding (*Figure 2—figure supplement 1B*) as previously observed (*Ishida et al., 2012*), but did not affect Suv39h1-CD's RNA binding (*Figure 2—figure supplement 1B,C*). These results suggested that Suv39h1-CD's ability to bind RNA is an intrinsic property that is distinct from its ability to recognize H3K9me3. Although addition of unmodified H3 peptide appeared to have a negative effect on the RNA-binding activity for both Chp1-CD and Suv39h1-CD (*Figure 2—figure supplement 1B*), it was presumably due to an electrostatic interaction between the unmodified H3 peptide and RNA probe.

We also investigated whether Suv39h1-CD's nucleic acid binding influences its recognition of methylated H3K9, even though its RNA binding was not affected by adding H3K9me3 peptide. We performed peptide pull-down assays in the presence or absence of major satellite ssRNA or dsDNA (*Figure 2—figure supplement 1D*). The presence of these nucleic acids slightly repressed rather than enhanced the overall amount of co-pulled-down Suv39h1-CD, suggesting that Suv39h-CD's recognition of methylated H3K9 is also independent of its nucleic acid binding. This slight reduction in Suv39h1-CD's H3K9me3 binding was probably due to an electrostatic interaction between the H3 peptide and the ssRNA/dsDNA probe. Furthermore, we confirmed that Suv39h1-CD's recognition of methylated H3K9 was not affected by the nucleic acid-binding defective mutation, 4A (*Figure 2C*, see below).

## Basic residues in Suv39h1-CD's C-terminal α-helix are involved in binding RNA

To understand the biological implications of Suv39h1-CD's RNA binding, we next investigated the residues required for binding RNA. The CD consists of three β-sheets and a C-terminal α-helix (*Figure 2A*) (*Jacobs and Khorasanizadeh, 2002*; *Nielsen et al., 2002*; *Wang et al., 2012*). Our previous studies of fission yeast Chp1-CD and Clr4-CD demonstrated that a stretch of basic residues in the C-terminal α-helix is required for their RNA binding (*Ishida et al., 2012*). Although Suv39h1-CD does not appear to have such a stretch of basic residues, it has a series of basic residues in its C-terminal α-helix (*Figure 2A*, indicated in red). We introduced five different amino acid substitutions

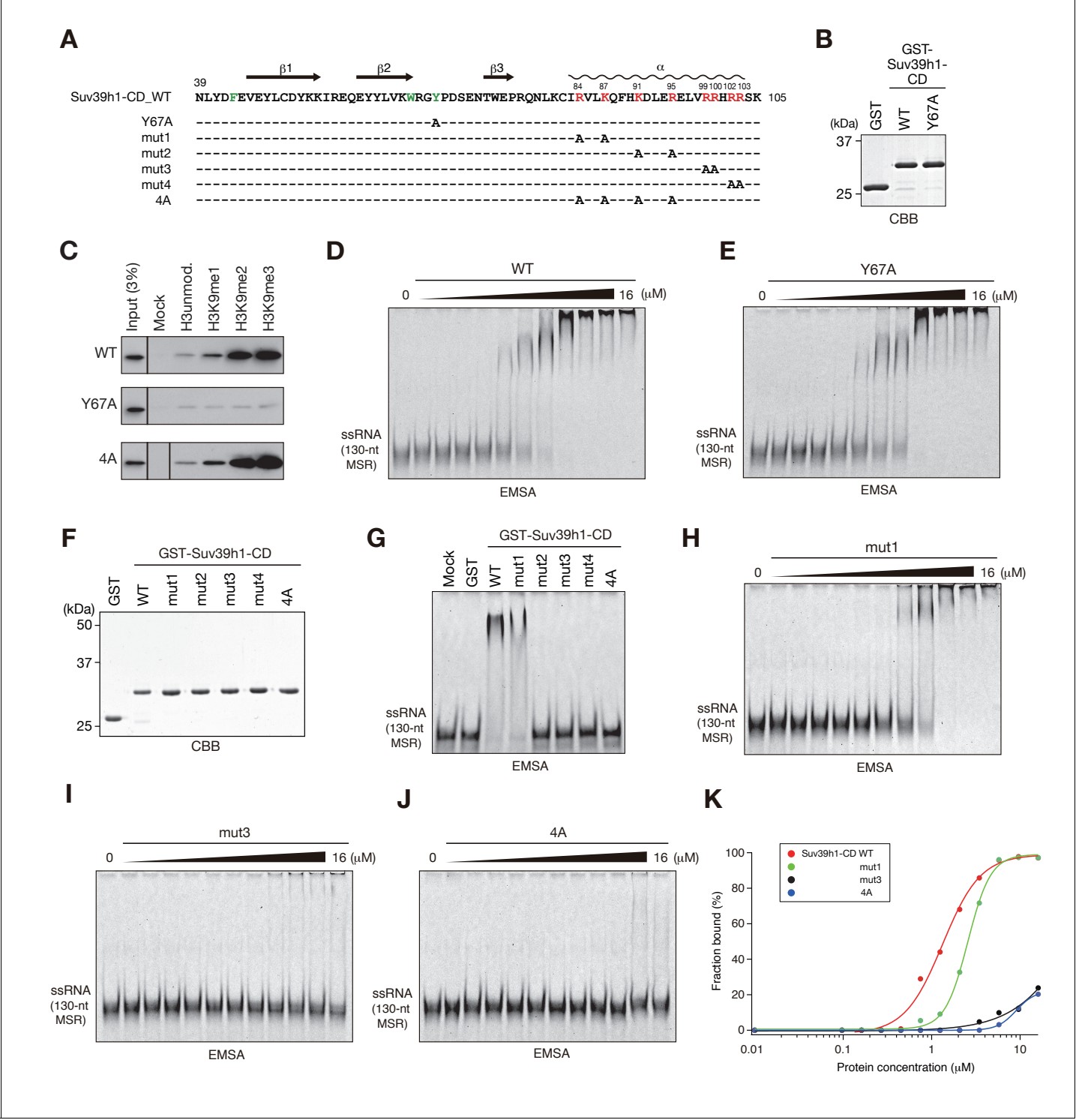

**Figure 2.** The residues required for Suv39h1-CD to bind RNA. (**A**) The alignment of wild-type (WT) and mutant Suv39h1-CD (39–105). The upper lines indicate secondary structure. The positively charged amino acids in the C-terminal α-helix are indicated in red. The aromatic cage residues responsible for Suv39h1-CD's recognition of H3K9me3 were indicated in green. (**B, F**) The recombinant proteins used in (**C–E and G–J**); the proteins were visualized by CBB staining. (**C**) An *in vitro* peptide-binding assay using wild-type (WT) and mutant (Y67A or 4A) Suv39h1-CDs. Biotin-tagged H3 (1–21) peptide (100 pmol) was incubated with GST-Suv39h1-CD (1 pmol), and the pulled-down proteins were analyzed by western blotting with an anti-GST antibody. (**D, E, H–J**) Titration EMSAs using serially diluted GST-fused WT or mutant Suv39h1-CD. (**G**) An EMSA using GST-fused WT or mutant Suv39h1-CD. (**K**) Binding isotherms of WT and mutant Suv39h1-CD proteins for ssRNA.

*Figure 2 continued on next page*

*Figure 2 continued*

The following figure supplement is available for figure 2:

**Figure supplement 1.** Suv39h1-CD's RNA binding is independent of its H3K9me3 recognition.

into Suv39h1-CD (*Figure 2A,F*; mut1–4 and 4A) and tested the ability of these mutants to bind RNA. In EMSAs, a Suv39h1-CD mutant (mut1) with N-terminal distal amino acid substitutions bound RNA more weakly than wild-type Suv39h1-CD; the mutants with other substitutions almost completely lost the ability to bind RNA (*Figure 2G*). Titration analyses confirmed that these basic residues were critical for Suv39h1-CD's RNA binding (*Figure 2H–K*). Our results demonstrated that the basic residues in Suv39h1-CD's C-terminal α-helix are also involved in its RNA binding as those in Chp1-CD or Clr4-CD are.

## Full-length Suv39h1 exhibits diminished nucleic-acid binding activity *in vitro*

Fission yeast Chp1 binds to RNA via both its N-terminal CD and its central RNA recognition motif (RRM), and the CD and RRM cooperate to repress centromeric transcripts (*Ishida et al., 2012*). To determine how other Suv39h1 domains contribute to its CD's RNA/DNA binding, we prepared Suv39h1-CD and full-length Suv39h1 as maltose binding protein (MBP)-fused versions (*Figure 1—figure supplement 1J*), since we noticed that the solubility of MBP-Suv39h1 was better than that of GST-Suv39h1 (data not shown). We first confirmed that MBP-Suv39h1-CD bound 130-nt ssRNA with a similar affinity to that of GST-Suv39h1-CD (*Figure 1D*, *Figure 1—figure supplement 1L*). Interestingly, we found that MBP-fused full-length Suv39h1 bound ssRNA with a relatively higher affinity compared with MBP-Suv39h1-CD, and the 4A mutation partially impaired this binding (*Figure 1—figure supplement 1L–O*). These results suggest that other Suv39h1 domain(s) contribute to Suv39h1-CD's RNA-binding activity, which was reminiscent of fission yeast Chp1 (*Ishida et al., 2012*).

## Both Suv39h1-CD's nucleic acid binding and methylated H3K9 recognition/binding contribute to Suv39h1's retention on pericentric heterochromatin

To elucidate how Suv39h-CD's nucleic acid–binding activity is integrated into the Suv39h1-mediated establishment and maintenance of pericentric heterochromatin, we prepared retroviral expression vectors expressing FLAG-tagged Suv39h1 mutants defective in nucleic acid binding (4A), methylated H3K9 binding (Y67A), or both (Y67A-4A), and FLAG-tagged wild-type Suv39h1 (*Figure 3A*). We transduced these constructs into *Suv39h* dn immortalized mouse embryonic fibroblasts (iMEFs) (*Lachner et al., 2001*). On day 2 after infection with these viral particles, the GFP-positive cells were sorted out to collect transgene-expressing *Suv39h* dn iMEFs. The sorted cells were analyzed for pericentric heterochromatin assembly on day 3 and day 6 (*Figure 3C–F* and *Figure 3—figure supplement 1A,B*). We monitored these cells' exogenously expressed wild-type and mutant Suv39h1 on day 8, and confirmed that their expression levels were comparable to or slightly higher than that of endogenous Suv39h1 (*Figure 3B*).

The induction of wild-type Suv39h1 (Suv39h1-WT) expression in *Suv39h* dn iMEFs rescued pericentric H3K9me3 signals in the DAPI-dense pericentric regions (H3K9me3 foci+), 64.0 ± 7.9% of the cells on day 3 and 97.0 ± 1.3% on day 6 were H3K9me3-positive (*Figure 3C,D*). Inducing the H3K9me3-binding-deficient Suv39h1-Y67A mutant also rescued the H3K9me3 signals on most of the DAPI-dense regions on day 6 (90.1 ± 3.2%). The formation of H3K9me3 foci on DAPI-dense regions was also induced by the RNA-binding-deficient Suv39h1-4A mutant, but this induction was significantly less efficient than that observed with Suv39h1 (40.3 ± 1.9% on day 3 and 87.3 ± 0.6% on day 6). The Suv39h1-Y67A-4A double mutant had an additive effect, and H3K9me3 foci+ cells were further decreased on day 6 (71.9 ± 1.9%). We further examined H3K9me3 status on pericentromere (major satellite repeat regions) by chromatin immunoprecipitation (ChIP) analysis. H3K9me3 level at major satellite regions was severely diminished in *Suv39h* dn iMEFs and recovered by FLAG-tagged Suv39h1-WT expression on day 6 after transduction (*Figure 3—figure supplement 1C*). Suv39h1

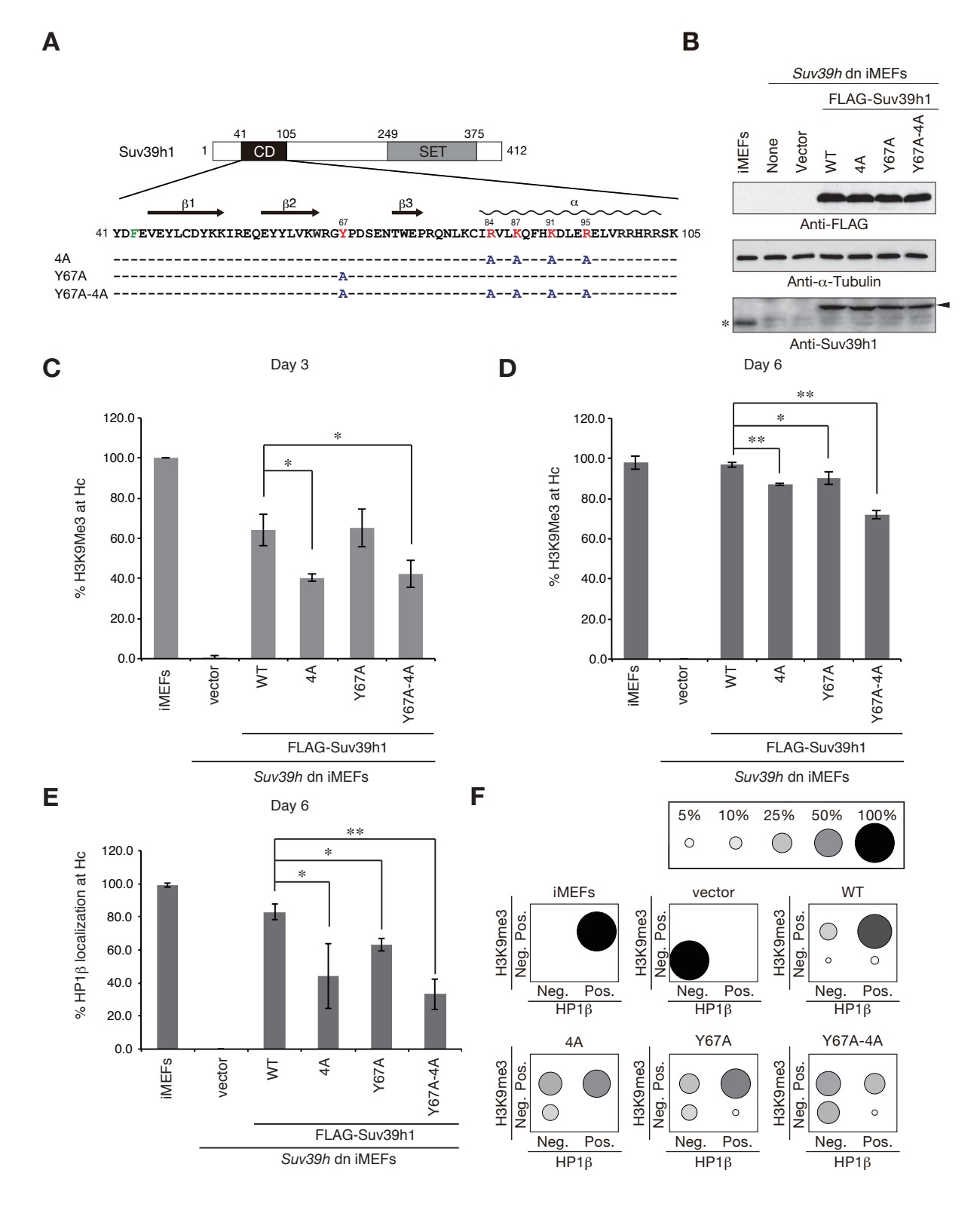

**Figure 3.** Suv39h1-CD's nucleic acid binding and H3K9me binding are both crucial for pericentric heterochromatin assembly. (**A**) Alignment of wild-type (WT) and mutant full-length Suv39h1. Boxes outline the chromodomain (shaded black), and the SET domain (shaded gray). (**B**) Suv39h1 protein expression level. *Endogenous Suv39h1; arrowhead: FLAG-Suv39h1. (**C–D**) Kinetic comparison of the appearance ratio of the cells containing H3K9me3 + DAPI-dense foci after induction of WT and mutant Suv39h1 in *Suv39h* dn iMEFs. Immunohistochemical staining for H3K9me3 in WT iMEFs, *Suv39h* dn

*Figure 3 continued on next page*

*Figure 3 continued*

iMEFs, or *Suv39h* dn iMEFs expressing FLAG-tagged 4A, Y67A, Y67A-4A, or WT Suv39h1 at (C) 3 days or (D) 6 days post-infection. The H3K9me3-positive cells within a population of 100 or more cells were counted (n = 3, mean ± SD, *p<0.05, **p<0.001). (E) The rate of HP1β heterochromatin accumulation in WT and mutant Suv39h1 during heterochromatin establishment. Immunohistochemical staining for HP1β in WT iMEFs, *Suv39h* dn iMEFs, or *Suv39h* dn iMEFs expressing FLAG-tagged 4A, Y67A, Y67A-4A, or WT Suv39h1 at 6 days post-infection. (F) HP1β accumulated at heterochromatin in cells where H3K9me3 was restored. The radii of the circles indicate the number of cells in each category. Immunohistochemical staining for HP1β in WT iMEFs, *Suv39h* dn iMEFs, or *Suv39h* dn iMEFs expressing FLAG-tagged 4A, Y67A, Y67A-4A, or WT Suv39h1 at 6 days post-infection. The cells with HP1β heterochromatin accumulation within a population of 100 cells or more were counted (n = 3, mean ± SD, *p<0.05, **p<0.005). The radii of the circles indicate the % ratio of cells in each category.

The following figure supplements are available for figure 3:

**Figure supplement 1.** H3K9me3 or HP1β in wild-type and mutant Suv39h1 during heterochromatin assembly.

**Figure supplement 2.** Comparable histone H3 methylation by wild-type and mutant Suv39h1 *in vitro*.

**Figure supplement 3.** Cell proliferation rates of *Suv39h* dn iMEFs were not changed by expressing FLAG-tagged WT or mutant Suv39h1.

**Figure supplement 4.** Suv39h1's N-terminal region also contributes to Suv39h1-mediated pericentric H3K9me3 formation.

mutants (Suv39h1-Y67A, −4A and -Y67A-4A) also increased H3K9me3 levels, but Suv39h1-Y67A-4A had a minimal effect. An *in vitro* HMTase assay showed that all of the Suv39h1 mutants had histone methyltransferase (HMTase) activity for free H3, comparable to Suv39h1-WT (*Figure 3—figure supplement 2*). Cell proliferation analysis demonstrated that the number of cell divisions during the first 3 days after viral infection were not much different among the *Suv39h1* dn iMEFs infected with an empty or expression vector for Suv39h1 proteins (*Figure 3—figure supplement 3*). Thus, it is unlikely that the different rescue efficiencies for H3K9me3 resulted from altered enzymatic activity or by cell cycle delay.

In the case of pericentric HP1β accumulation, Suv39h1-WT expression induced >80% of cells to become HP1β foci-positive (HP1β foci+) on day 6 (83.0 ± 4.9%), but the phenotype was rescued only poorly by the Suv39h1 mutants (*Figure 3E*; 63.0 ± 3.7%, 44.3 ± 19.8%, and 33.2 ± 9.4% for Suv39h1-Y67A, −4A, and -Y67A-4A on day 6, respectively). Dual immunostaining of the *Suv39h* dn iMEFs on day 6 further demonstrated that compared to iMEFs rescued with Suv39h1-WT, iMEFs rescued with the Suv39h1 mutants contained larger cell populations that were negative for both H3K9me3 and HP1β (*Figure 3F*, bottom left of each panel) or were positive for only H3K9me3 (*Figure 3F*, top left of each panel), suggesting that SUV39H-mediated heterochromatin assembly was less efficient in those mutant cells.

To examine whether introduced Suv39h1 actually restored the repressive heterochromatin structure, we analyzed the expression levels of major satellite RNA. As reported previously (*Bulut-Karslioglu et al., 2012*), major satellite repeats were derepressed in *Suv39h* dn iMEFs (~10 times higher than that of WT iMEFs) and introduction of Suv39h1-WT suppressed this up-regulation of major satellite repeats (*Figure 3—figure supplement 1D*). Mutant Suv39h1 with Y67A, 4A, or Y67A-4A also decreased major satellite repeat transcripts, but their repression levels were milder than that of Suv39h1-WT. Furthermore, among these mutants, Suv39h1-Y67A-4A's effect was the weakest. These results are well consistent with the restored levels of H3K9me3 and HP1β.

Collectively, these results indicate that Suv39h1-CD's nucleic acid–binding activity integrates and cooperates with its recognition and binding of methylated H3K9 to establish/maintain Suv39h1-mediated pericentric heterochromatin such as H3K9me3 formation, HP1 accumulation, and transcriptionally repressive status.

## Suv39h1's N-terminal region also contributes to Suv39h1-mediated pericentric H3K9me3 formation

Despite having impaired binding of both RNA and methylated H3K9, the Suv39h1 mutant Y67A-4A was able to assemble CH on pericentric regions, although less efficiently and completely than wild-type Suv39h1 (*Figure 3*). These data indicate that the Y67A-4A mutant retains other critical factors

or functions that promote Suv39h1's association with CH sufficiently for Y67A-4A to induce pericentric heterochromatin assembly. As to which factor or function is involved in this process, HP1 or an HP1-associated function is a reasonable candidate, since Suv39h1's N terminus interacts with HP1 (*Melcher et al., 2000*; *Yamamoto and Sonoda, 2003*). Indeed, we recently found that the induction of pericentric H3K9me3 is extremely poor in *Suv39h* dn iMEFs complemented with an N-terminal deletion (Δ1–41) mutant of Suv39h1 (ΔN, *Figure 3—figure supplement 4A,C,D*) expressed at an endogenous level (*Muramatsu et al., 2016*) (*Figure 3—figure supplement 4B*). Although ΔN does not bind HP1, it was recently reported that the N-terminal region of Suv39h1 (1-41) also possesses chromatin-binding activity independent of its HP1 interaction (*Müller et al., 2016*). Therefore, we validated the role of this chromatin-binding activity in Suv39h1-mediated H3K9me3 formation. We used the Suv39h1-R24,K27A mutant, which is reported to be defective for chromatin binding but to still interact with HP1 in pericentric H3K9me3 formation in *Suv39h* dn iMEFs (*Müller et al., 2016*) (*Figure 3—figure supplement 4B–D*). We found that the pericentric H3K9me3 formation was clearly diminished in *Suv39h* dn cells complemented with Suv39h1-R24,K27A, but the impact of ΔN was much stronger than that of the R24,K27A mutation. Thus, we concluded that both the HP1-binding and chromatin-binding activities of the N-terminal region also contribute to Suv39h1-mediated heterochromatin assembly.

## Suv39h1-CD's nucleic acid binding is required for its interaction with major satellite RNAs *in vivo*

Since Suv39h1-CD had a higher binding affinity for RNA than for DNA in EMSAs (*Figure 1D–G*), and this binding activity appeared to be crucial for Suv39h1-mediated pericentric heterochromatin formation, we examined whether Suv39h1 binds RNA transcribed from pericentric major satellite repeats inside the cell. In experiments with *Suv39h* dn iMEFs expressing FLAG-tagged wild-type or 4A Suv39h1, we immunoprecipitated cellular RNAs associated with the wild-type or mutant Suv39h1 and assessed the levels of the major satellite repeats by dot-blot hybridization, which allowed us to evaluate the levels of repetitive DNA without amplification. As shown in *Figure 4*, the signal for the major satellite repeats was clearly detected in precipitates from iMEFs expressing wild-type Suv39h1 but was severely diminished in precipitates from iMEFs expressing mutant Suv39h1-4A (*Figure 4B*), even though similar amounts of FLAG-Suv39h1-WT or −4A were immunoprecipitated (*Figure 4A*). These major satellite signals were totally lost by pretreating the samples with RNase, indicating that the detected signals were derived from RNA, not from contaminating genomic DNA. Although our EMSA using full-length Suv39h1 suggested the involvement of Suv39h1 domains other than CD in RNA binding (*Figure 2—figure supplement 1J–O*), these data show that Suv39h1 binds major satellite RNA primarily via its CD. The major satellite repeats are reported to be expressed during G1/S transition and in M phase (*Lu and Gilbert, 2007*). Therefore, this expression during G1/S transition may have a role in H3K9me3 inheritance after replication.

The interaction between Suv39h1 and major satellite RNAs was also examined by RT-qPCR analysis (*Figure 4—figure supplement 1*). Suv39h1-WT stably bound major satellite RNAs, and this interaction was impaired for Suv39h1-4A. The interaction was also lost for Suv39h1-Y67A mutant, suggesting that H3K9me3 recognition by Suv39h1-CD is also required for the Suv39h1's stable interaction with the major satellite RNAs. Intriguingly, Suv39h1-Y67A-4A interacted partially with the major satellite RNAs. This result does not seemingly fit with our model that Suv39h1-CD's RNA- and H3K9me3-binding activities cooperatively function in Suv39h1's chromatin targeting. However, we also found that Suv39h1-Y67A-4A loosely bound chromatin (see below). Considering this observation, Suv39h1-Y67A-4A may interact cellular RNAs without stable binding to chromatin.

To determine whether the interaction between Suv39h1 and major satellite RNA occurs in the nucleus, we conducted chromatin fractionation assays. Cells were harvested and separated into soluble (Sup) and insoluble, chromatin-enriched (Ppt) fractions, and endogenous Suv39h1's chromatin association was evaluated by immunoblotting. We found that most of Suv39h1 was present in the chromatin-enriched fraction (*Figure 4C*). We confirmed that FLAG-tagged wild-type Suv39h1 expressed in *Suv39h* dn iMEFs was also associated with chromatin (*Figure 4D*). Interestingly, although Suv39h1 mutants (4A and Y67A) failed to interact with major satellite RNA in the RNA-IP experiments (*Figure 4B* and *Figure 4—figure supplement 1*), most of these mutants were also detected in the chromatin-enriched fractions (*Figure 4D*), suggesting that Suv39h1 mutants are still associated with chromatin even without being stably bound to major satellite RNA. Even more

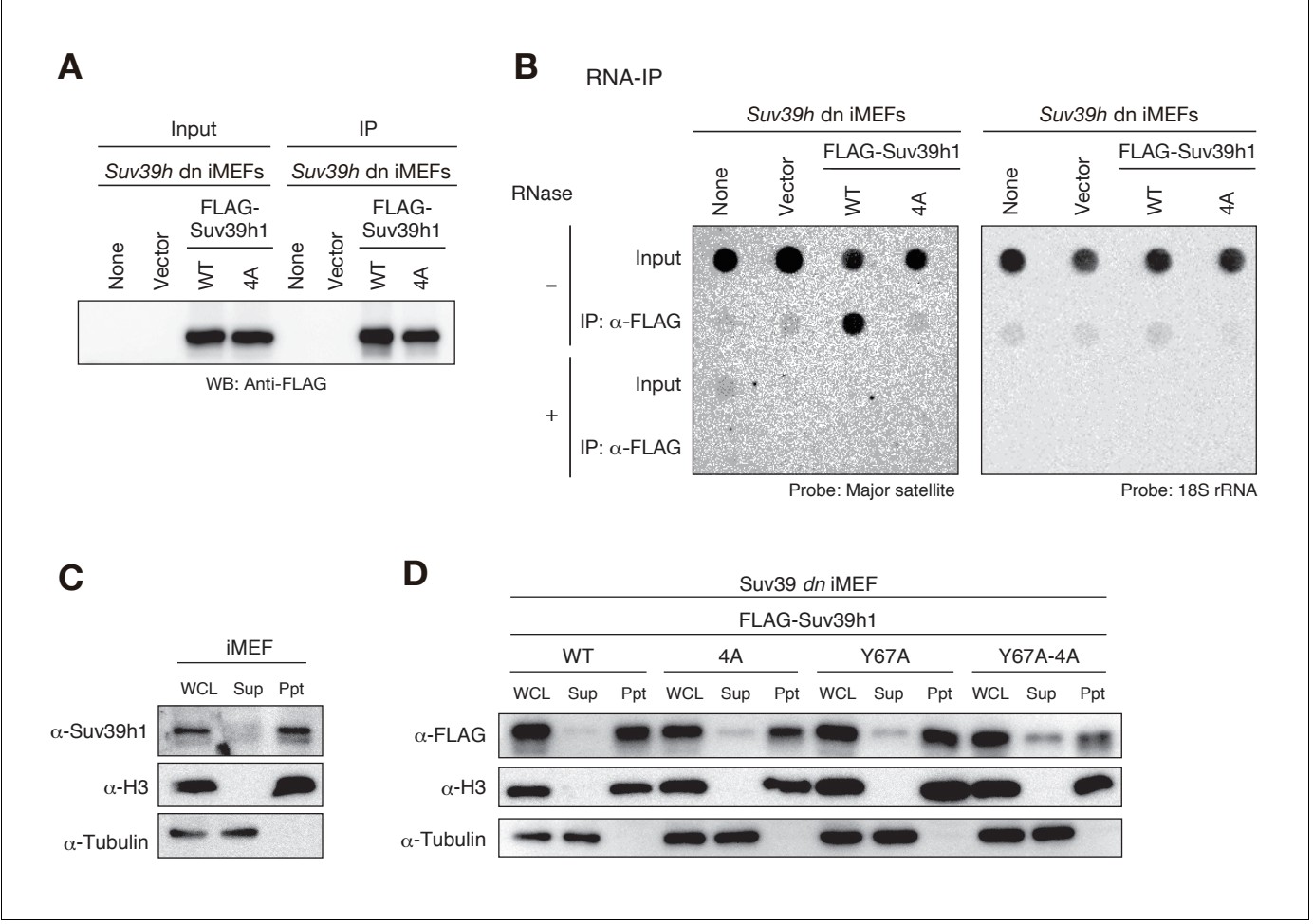

**Figure 4.** Suv39h1-CD's nucleic acid binding is required for its interaction with major satellite RNAs in vivo. (**A**) Whole cell lysates (input) of *Suv39h* dn iMEFs expressing FLAG-tagged wild-type (WT) or mutant (4A) Suv39h1 and FLAG immunoprecipitates (IP) were subjected to immunoblotting using an anti-FLAG M2 antibody. Parental *Suv39h* dn iMEFs (none) and *Suv39h* dn iMEFs with an empty vector (vector) were used as controls. (**B**) Dot-blot analysis of immunoprecipitated RNAs. RNAs associated with WT or mutant (4A) Suv39h1 in *Suv39h* dn iMEFs were precipitated with the anti-FLAG M2 antibody and subjected to dot-blot analysis using a labeled probe for major satellite repeats (left) and 18S rRNA (right). (**C, D**) Chromatin fractionation assays were performed using control iMEFs (**C**) and *Suv39h* dn iMEFs expressing FLAG-tagged WT or mutant Suv39h1 (**D**). Soluble (Sup) and insoluble chromatin-enriched (Ppt) fractions were resolved by SDS-PAGE and analyzed by immunoblotting. For comparison, anti-α-tubulin and anti-histone H3 antibodies were used to detect soluble- and chromatin-associated proteins, respectively.

The following figure supplement is available for figure 4:

**Figure supplement 1.** Both Suv39h1-CD's nucleic acid- and H3K9me3 binding activities are required for its interaction with major satellite RNAs *in vivo*.

interestingly, a relatively higher level of Suv39h1-Y67A-4A was detected in the soluble fractions (*Figure 4D*). Together, these results confirmed that Suv39h1 interacts with major satellite RNA within the nucleus, but not in the cytoplasm, and that both RNA- and H3K9me3-binding activities contribute to Suv39h1's chromatin association.

## Both the nucleic acid and methylated H3K9 recognition/binding activities of Suv39h1-CD impact Suv39h1 dynamics in pericentromeric regions

Our rescue experiments of *Suv39h* dn iMEFs clearly indicated that Suv39h1-CD's H3K9me3- and RNA-binding functions are both critical for establishing H3K9me3 marks in pericentric regions without preexisting H3K9me3 marks. On the other hand, we were still able to induce pericentric

H3K9me3 and HP1 accumulation in *Suv39h* dn iMEFs complemented with 4A, Y67A, or Y67A-4A Suv39h1 mutants (*Figure 3*), albeit less efficiently than in iMEFs expressing wild-type Suv39h1. Thus, Suv39h1-CD's function might be dispensable for maintaining H3K9me3 marks once established. However, Suv39h1-4A's association with major satellite RNA was severely impaired in cells with rescued H3K9me3 and HP1 (*Figure 4*), suggesting that defective RNA binding may affect Suv39h1 dynamics in the pericentromere, where major satellite repeat elements are transcribed. Consistent with this possibility, a single K87A amino acid substitution in SUV39H1 (included in mut1 and 4A in our study) was reported to enhance the SUV39H1 exchange rate in CH regions (*Bosch-Presegué et al., 2011*), although the biological implication of this observation remains unclear. It is quite possible that Suv39h1 dynamics in the pericentromere were also affected in the mutant with defective H3K9me3 binding.

To test the possibility that Suv39h1's RNA- and H3K9me3-binding contribute to its dynamics on CH, we measured these dynamics in *Suv39h* dn iMEFs with stable expression of GFP-tagged wild-type or mutant Suv39h1 (*Figure 5—figure supplement 1A*). As expected, GFP-Suv39h1-WT localized to the nucleus and formed foci on Hoechst-dense pericentric regions (*Figure 5A*). GFP-Suv39h1-4A, -Y67A, and -Y67A-4A also formed foci on Hoechst-dense regions (*Figure 5A*). However, semi-quantitative analysis to measure their heterochromatin enrichment (represented as the intensity ratio of euchromatin over heterochromatin; *Figure 5B*) revealed that GFP-Suv39h1-Y67A-4A fluorescence was significantly increased in euchromatin compared to the wt (*Figure 5B*), suggesting that Suv39h1-Y67A-4A's affinity to heterochromatic regions was decreased. GFP-Suv39h1-Y67A was also slightly less enriched on heterochromatin, and Y67A and 4A appeared to have an additive effect. We next assayed these molecules' binding time to heterochromatin by fluorescence recovery after photobleaching (FRAP). As clearly shown in *Figure 5C and D*, GFP-Suv39h1-4A and -Y67A exchanged more rapidly than Suv39h1-WT (GFP-Suv39h1-4A and -Y67A had recovery half-times of ~6 s, versus ~20 s for Suv39h1-WT), and their mobile fraction was increased (88–89% for GFP-Suv39h1-4A and -Y67A, versus ~73% for GFP-Suv39h1-WT). GFP-Suv39h1-Y67A-4A was much more dynamic, with an ~3 s recovery half-time and ~91% mobile fraction. GFP-Suv39h1-mut1 and -mut3 (*Figure 5—figure supplement 1*) also exchanged more rapidly than GFP-Suv39h1-WT (*Figure 5—figure supplement 1C,D*). These data suggest that Suv39h1-CD's binding of nucleic acid and of methylated H3K9 are both crucial for Suv39h1's retention of heterochromatin in cells.

## Major satellite RNAs contribute to retention of Suv39h1 on heterochromatin

Since the Suv39h1-4A mutant is defective in major satellite RNA binding and more dynamic on CH *in vivo*, it is highly possible that major satellite repeat transcripts contribute to the retention of Suv39h1 on heterochromatin. However, the above experiments could not rule out the possibility that Suv39h1-CD targets chromatin via its DNA-binding activity (*Figure 1E*). We therefore employed knock-down of major satellite repeat transcripts using locked nucleic acid (LNA)-DNA gapmers (*Probst et al., 2010*). As shown in *Figure 6A*, introduction of major satellite repeat-specific LNA-DNA gapmers into *Suv39h* dn iMEFs that express GFP-Suv39h1 significantly and specifically decreased major satellite repeat transcripts. In major satellite knock-down cells, GFP-Suv39h1 on pericentric heterochromatin became more mobile than the control (*Figure 6B*). This data further supports the notion that major satellite RNAs contribute to Suv39h1 stability on pericentric regions.

## Discussion

The CD is a highly conserved protein module most recognized for targeting methylated histone tails (*Blus et al., 2011*; *Fischle et al., 2003*). Although several lines of evidence suggest that the CD binds DNA and RNA in addition to targeting methylated histones, its biological relevance and physiological roles are poorly understood (*Hiragami-Hamada and Fischle, 2014*). In this study, we demonstrated that Suv39h1-CD not only recognizes H3K9me3, but also binds RNA and DNA (with a higher affinity to ssRNA than to dsDNA), indicating that RNA may also be crucial for Suv39h1 recruitment/function. Indeed, our current data suggest that these two activities contribute to Suv39h1-mediated CH assembly, including the accumulation of H3K9me3 and HP1 (*Figure 3*) and the retention of Suv39h1 itself on the pericentric regions (*Figure 5*). Furthermore, we found that the HP1-binding and chromatin-binding activities of Suv39h1's N-terminal region contributed to the Suv39h1-

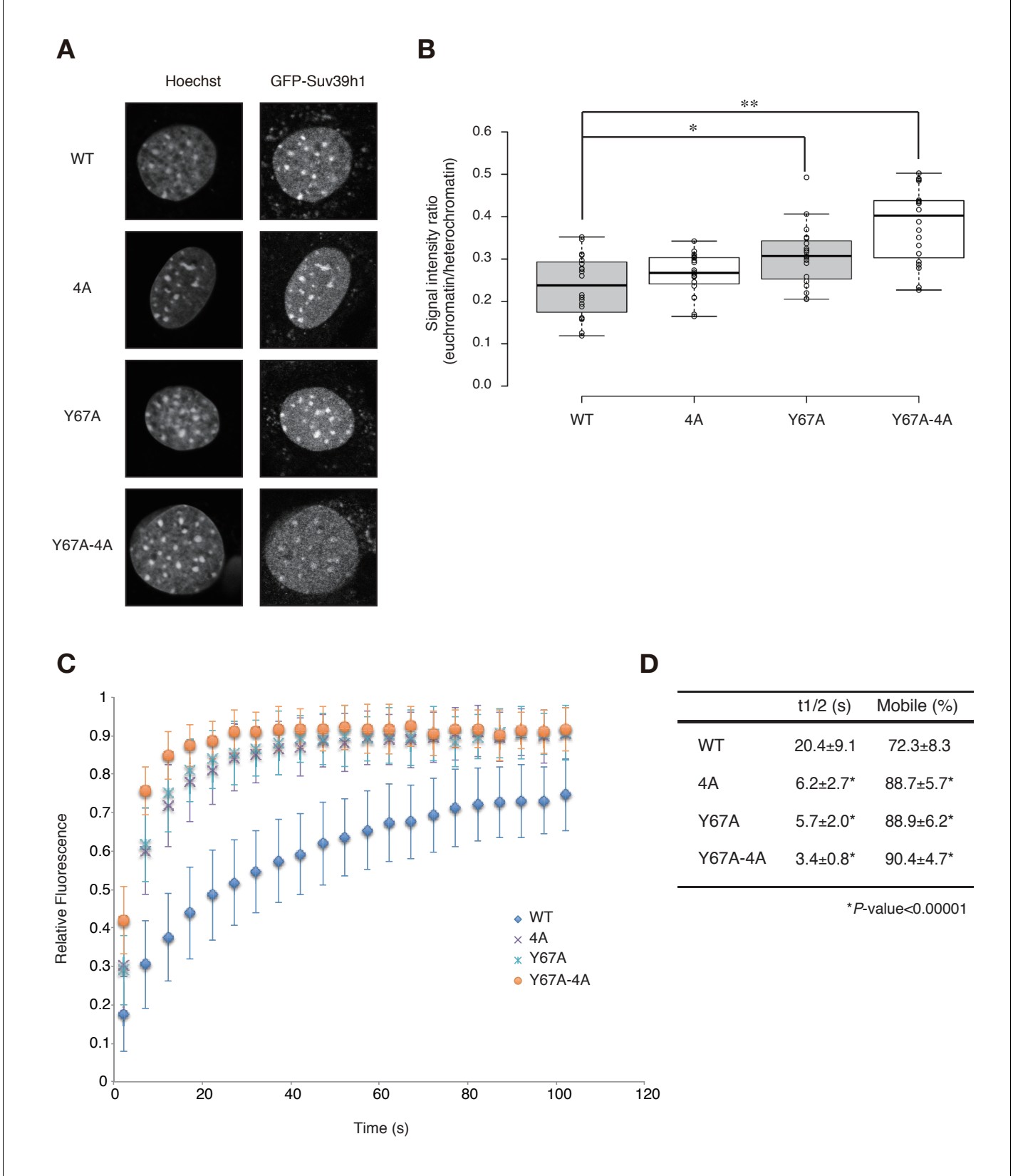

**Figure 5.** Both Suv39h1-CD's nucleic acid binding and H3K9me binding contribute to Suv39h1's retention on heterochromatin. (**A**) Subcellular localization of Hoechst staining and wild-type (WT) or mutant GFP-Suv39h1 in *Suv39h* dn iMEF cells. (**B**) Box plot showing the signal-intensity ratio

*Figure 5 continued on next page*

*Figure 5 continued*
between euchromatin and heterochromatin. Center lines show the medians, box limits indicate the 25th and 75th percentiles as determined by R software, whiskers extend 1.5 times the interquartile range from the 25th and 75th percentiles, and outliers are represented by dots; n = 20 sample points. (*p<0.005, **p<0.00005) (C) FRAP analysis of WT or mutant GFP-Suv39h1 in *Suv39h* dn iMEFs. The means of the relative intensity in the bleached area are indicated with the SD (n ≥ 29). (D) Quantification and statistical analysis of the FRAP analysis in C; and the mobile population fraction (mobile) and half-time of fluorescence recovery (t1/2).
The following figure supplement is available for figure 5:

**Figure supplement 1.** Wild-type and mutant GFP-Suv39h1 analysis.

mediated pericentric H3K9me3 formation (*Figure 3—figure supplement 4C* and [*Muramatsu et al., 2016*]). Collectively, our results demonstrated that the four different activities of Suv39h1 other than its lysine methyltransferase activity cooperatively regulate Suv39h1-mediated pericentric heterochromatin assembly. Independent work by another group shows that *in vitro* RNA binding activity is also associated with the CD of human SUV39H1 (*Johnson et al., 2017*).

We found that basic residues in Suv39h1-CD's C-terminal α-helix are involved in binding RNA, and that Suv39h1-CD's RNA binding is distinct and independent of its recognition and binding of H3K9me3. Considering the slower recovery of H3K9me3 in *Suv39h* dn iMEFs expressing Suv39h1-4A (which is deficient in RNA binding) than in those expressing Suv39h1-Y67A (which is deficient in H3K9me3 binding), RNA-binding activity is particularly involved in recruiting Suv39h1 to H3K9me3-free chromosomal regions. Since pericentric regions consist of major satellite repeats, it is possible that nascent transcripts derived from such regions possess specialized properties that are recognized by Suv39h1. Although SUV39H1 is reported to preferentially bind TERRA transcripts derived from telomeric repeats (*Porro et al., 2014*), our study demonstrates that Suv39h1-CD also efficiently binds *ACTB* mRNA and does not show any particular sequence specificity (*Figure 1—figure supplement 1*). Thus, it is unlikely that Suv39h1 targets specific chromosomal regions by recognizing a specific RNA sequence or structure. However, it is possible that transcripts derived from repetitive chromosomal regions are not efficiently processed, leading to their stable association with transcribed regions. Pericentric transcripts with this kind of feature might become a target of Suv39h1. Since another independent work now demonstrates that the N-terminal basic domain of mouse Suv39h2 also shows RNA binding activity and this activity is crucial for the recruitment and stabilization of Suv39h2 to heterochromatin. (*Velazquez Camacho et al., 2017*), Suv39h-family proteins may use a similar RNA binding activity for their targeting.

In addition to its role in establishing H3K9me3, Suv39h1-CD's RNA-binding activity also contributes to Suv39h1's stable association with H3K9me3-rich pericentromeric regions (*Figure 5*). Knockdown of major satellite repeat transcripts (*Figure 6*) further supports the notion that RNA (major satellite RNA) binding of Suv39h1 contributes to its stability on chromatin.

While we showed that Suv39h1-CD binds H3K9me3 peptide, its binding affinity is weaker than those of the CDs of HP1 proteins (*Figure 2—figure supplement 1A* and [*Jacobs and Khorasanizadeh, 2002*; *Wang et al., 2012*]). Suv39h1-CD's RNA-binding activity might increase its affinity for H3K9me3-rich pericentric regions; alternatively, RNA binding might modulate Suv39h1-CD's enzymatic activity. A recent report showed that Suv39h1's CD inhibits its methyltransferase activity, and the CD's binding of H3K9me3 is suggested to allosterically regulate Suv39h1's enzymatic activity (*Müller et al., 2016*). It is not clear how Suv39h1's methyltransferase activity is controlled by the CD; however, it is possible that Suv39h1-CD's RNA binding is also coupled with its allosteric regulation. Physical interaction between the CD and SET domains appeared to affect the CD's RNA-binding activity: we demonstrated that full-length Suv39h1 bound RNA with a relatively higher affinity compared with Suv39h1-CD (*Figure 1—figure supplement 1M*). Considering the CD's role in targeting Suv39h1 to H3K9me3-free chromatin regions, Suv39h1's physical association with other factors such as chromatin-unbound non-coding RNAs or HP1 proteins may modulate the CD's C-terminal α-helix to bind RNAs.

In general, the assembly of higher-order chromatin structure has two steps, establishment and maintenance (*Hall et al., 2002*; *Sadaie et al., 2004*). H3K9me3 marks on CH have to be maintained during DNA replication, and given that Suv39h1 possesses a CD that recognizes H3K9me3, a direct

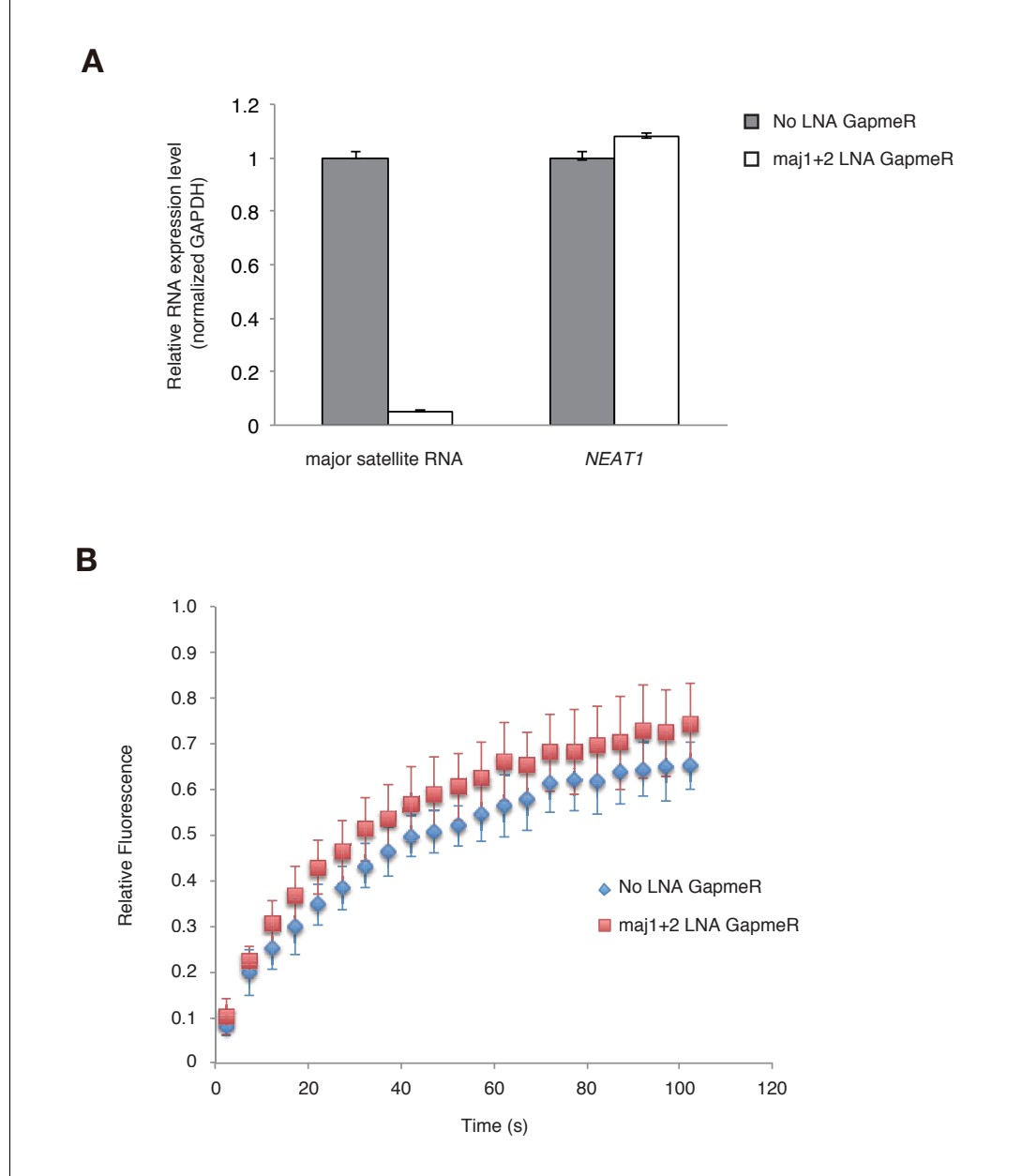

**Figure 6.** Major satellite RNAs contribute to Suv39h1's retention on heterochromatin. (**A**) qRT-PCR for major satellite RNAs or nuclear non-cording RNA *NEAT1* transcripts in *Suv39h* dn iMEFs expressing GFP-suv39h1 transfected without or with a set of two LNA-DNA GapmeRs directed against the Forward and Reverse major satellite transcripts. (**B**) FRAP analysis of GFP-Suv39h1 in the described cells (**A**). The means of the relative intensity in the bleached area are indicated with the SD (n ≥ 13).

read-write mechanism involving Suv39h1 is proposed to maintain H3K9me3 (*Audergon et al., 2015*; *Ragunathan et al., 2015*). However, this mechanism does not fully explain the restoration of H3K9me3 in *Suv39h* dn iMEFs. Thus, Suv39h1-CD's RNA-binding activity appears to contribute to its ability to target pericentric regions without preexisting H3K9me3 and to establish H3K9me3 marks; this function is consistent with the role of nucleic acid binding in Chp1 and Clr4 in fission yeast (*Ishida et al., 2012*). Moreover, HP1 can also bind RNA and methylated H3K9 (*Keller et al., 2012*; *Muchardt et al., 2002*). HP1's ability to bind RNA is SUMO-dependent, and an HP1-Ubc9 fusion molecule with enhanced sumoylation on the HP1 part can be targeted to the pericentric regions, even in *Suv39h* dn iMEFs (*Maison et al., 2011*). Thus, HP1 can recruit mutant Suv39h1 to pericentric

regions via binding between sumoylated HP1 and chromatin-bound major satellite RNA. Once Suv39h1-Y67A-4A is targeted to pericentric regions and deposits H3K9me3, HP1 further increases its affinity for pericentric regions, and this positive feedback loop between HP1 and Suv39h1-Y67A-4A may be sufficient for this mutant Suv39h1 to mediate pericentric heterochromatin assembly. The ΔN phenotype supports this possibility.

In this study we demonstrated that Suv39h1-CD's RNA-binding activity is required for its targeting to and assembly of heterochromatin, highlighting the functional role of pericentromeric transcripts in introducing H3K9me3 marks. Recent studies using fission yeast demonstrated that H3K9me3 can be maintained during DNA replication as a self-perpetuating epigenetic mark (*Audergon et al., 2015*; *Ragunathan et al., 2015*). Interestingly, the maintenance of H3K9me3 requires intact Clr4-CD. Epe1, a protein containing a jmjC-domain, counteracts this process by directly removing the H3K9me3 mark. Such counteracting enzymatic activities appear to be important for preventing the inappropriate inactivation of euchromatic genes by stochastically introduced H3K9me3 marks. Although it is not clear whether similar counteracting demethylase activities play a role in maintaining genome integrity in mammalian cells, pericentromeric transcripts may modulate such counteracting enzymatic activity to directly regulate chromatin dynamics.

## Materials and methods

### Electrophoretic mobility-shift assays (EMSAs)

EMSAs were performed as described previously (*Ishida et al., 2012*) using a PCR-amplified major satellite sequence as a dsDNA probe. For RNA probes, DNA fragments (major satellite, *ACTB* ORF, and alfa satellite) were PCR-amplified with a T7 promoter sequence, and the resultant PCR fragments were used for *in vitro* transcription by Thermo T7 RNA polymerases (Toyobo). Transcribed RNAs were purified on denaturing 7 M Urea Polyacrylamide Gel and extracted with Biomasher I (Assist). Purified ssRNAs and dsDNAs were labeled with the 5' EndTag Nucleic Acid Labeling System (Vector Laboratories, Cat# MB-9001, RRID:AB_2336073), and the labeled probes were incubated with GST- or MBP-fused proteins in 10 µl of binding buffer (20 mM HEPES-KOH pH 7.6, 100 mM KCl, 0.01% NP-40, and 1 mM dithiothreitol [DTT]) containing 4 U RNaseOUT (Invitrogen). The fluorescent dye can be conjugated with dsDNA at the single or both 5'-ends, and the number of conjugated dye could affect the probe migration in gels, resulting in appearance of doublet bands. Binding reactions were carried out on ice for 30 min. For titration EMSAs, serially diluted proteins (diluted 3:5 or 1:2) were used. To examine the relationship between RNA binding and H3K9me3 recognition, an equimolar amount of H3K9me3 peptide was added to the binding reactions. The reaction samples were loaded onto a 5% polyacrylamide gel with $0.5 \times$ Tris-borate EDTA buffer (TBE), and the labeled probes were detected with a Typhoon 9400 (GE Healthcare). The unbound fractions were measured with ImageMaster 1D software (GE Healthcare). Igor Pro software (WaveMetrics, RRID:SCR_000325) was used for curve fitting.

### Cells and antibodies

Eco-Pack 2–293 cells (Clontech) and 293T cells (gifted from Dr. Yao [*Ito et al., 2001*]) were grown in Dulbecco's Modified Eagle's Medium supplemented with 10% fetal calf serum. Wild-type and *Suv39h* dn iMEFs (*Lachner et al., 2001*) were grown in Dulbecco's Modified Eagle's Medium supplemented with 10% fetal calf serum, 0.1 mM beta-mercaptoethanol, and 1x non-essential amino acids. The following antibodies were used in this study: anti-FLAG-M2 (F3165: Sigma-Aldrich, RRID:AB_259529), anti-FLAG-M2-HRP (A8592: Sigma-Aldrich, RRID: AB_439702), anti-α-tubulin (T5168: Sigma, RRID: AB_477579), anti-Suv39h1 (8729: Cell Signaling, RRID: AB_10829612), anti-H3 (ab21054: Abcam, RRID: AB_880437), anti-H3K9me3 (ab8898: Abcam, RRID: AB_306848 and 2F3 (RRID: AB_2616099)(*Chandra et al., 2012*), anti-HP1β (BMP002: MBL, RRID: AB_843158), and anti-GST (27-4577-01: GE Healthcare, RRID: AB_771432).

### Plasmid construction

DNA fragments encoding Suv39h1-CD (residues 39–105 or 39–96) were PCR-amplified from pCAG-IRES-Puro-FLAG-Suv39h1 (*Muramatsu et al., 2016*). The amplified Suv39h1-full length or -CD was cloned into the TOPO pCRII vector (Clontech) and subcloned into the bacterial expression vector

pGEX-6P-3 (GE Healthcare) to make the GST-fusion molecule. The GST-Suv39h1-CD (39–96) was only used for ITC. To express FLAG-tagged wild-type or mutant Suv39h1, DNA fragments encoding FLAG-Suv39h1 were generated by PCR using pCAG-IRES-Puro-FLAG-Suv39h1 as the template, and the amplified FLAG-Suv39h1 was cloned into the TOPO pCRII vector. Suv39h1 mutagenesis was performed with the TOPO pCRII vector according to the manufacturer's instructions (Stratagene). Next, the wild-type or mutant FLAG-Suv39h1 was subcloned into the pMCs-IRES-GFP retrovirus expression vector (Cell Biolabs). To express GFP-Suv39h1 fusion molecules, the PCR-amplified DNA of wild-type or mutant Suv39h1 was subcloned into the pEGFP-C1 vector (Addgene), and the GFP-Suv39h1 fusion part was amplified again by PCR and subcloned into the BamH1 and Sall sites of pMCs-IRES-GFP from which IRES and the vector's own GFP sequences were deleted.

### Retrovirus production and cell transduction

For retrovirus production, Eco-Pack 2–293 cells (Clontech) were transfected with pMCs-FLAG-Suv39h1$^{WT/mut}$-IRES-GFP or pMCs-GFP-Suv39h1$^{WT/mut}$. The resultant retrovirus supernatants were used to infect control or *suv39h dn* iMEF cells (*Peters et al., 2001*). To obtain populations expressing similar levels of wild-type or mutant FLAG-Suv39h1 or GFP-Suv39h1, infected cells expressing the same levels of GFP were sorted by fluorescence-activated cell sorting (FACS).

### Immunofluorescence microscopy

Cells cultured on chamber slides (Matsunami Glass, SCS-008) were fixed in 4% paraformaldehyde for 8 min at room temperature. The fixed cells were permeabilized with 0.1% Triton X-PBS for 15 min at room temperature and blocked with 3% BSA in 0.1% Tween-20 TBS (3% BSA-TTBS) for 15 min at room temperature. Next, the cells were incubated with primary antibodies diluted in 3% BSA-TTBS for 1 hr at room temperature, washed twice with PBS, and incubated with secondary antibodies conjugated with Alexa568, 647 Fluor (Invitrogen) diluted in 3% BSA-TTBS for 1 hr at room temperature. The nuclei were counterstained with DAPI and observed under a confocal microscope (Olympus, FV1000) at the Support Unit for Bio-Material Analysis in the RIKEN BSI Research Center (RRC).

### RNA immunoprecipitation (RIP)

iMEFs in eight 150 mm dishes (~80% confluent) were fixed with 1% formaldehyde for 10 min at room temperature. The harvested cells were lysed in SDS lysis buffer (50 mM HEPES-KOH pH 7.6, 1% SDS, and 10 mM EDTA), and the cleared cell lysates were sonicated with a Bioruptor (Cosmo Bio). The samples were diluted with RIP dilution buffer (50 mM HEPES-KOH pH 7.6, 167 mM NaCl, 0.01% SDS, 1.1% Triton X-100, and 5 mM EDTA) supplemented with 40 U/ml RNase inhibitor (RNaseOUT, Invitrogen) and were immunoprecipitated with anti-FLAG M2 antibody–conjugated Dynabeads (Invitrogen) at 4°C overnight. The beads were washed with ice-cold low-salt buffer (20 mM HEPES-KOH pH 7.6, 150 mM NaCl, 0.1% SDS, 1% Triton X-100, 2 mM EDTA, and 20 U/ml RNaseOUT), high-salt buffer (20 mM HEPES-KOH pH 7.6, 500 mM NaCl, 0.1% SDS, 1% Triton X-100, 2 mM EDTA, and 20 U/ml RNaseOUT), LiCl buffer (10 mM HEPES-KOH pH 7.6, 0.25 M LiCl, 1% Nonidet P-40, 1% sodium deoxycholate, 1 mM EDTA, and 20 U/ml RNaseOUT), and diethylpyrocarbonate (DEPC)-treated water. The protein–RNA complexes were eluted with RIP elution buffer (10 mM HEPES-KOH pH 7.6, 300 mM NaCl, 0.5% SDS, 5 mM EDTA, and 40 U/ml RNaseOUT) and treated with 0.8 μg/ml Proteinase K for 1 hr at 42°C, followed by de-crosslinking at 65°C for at least 4 hr. Immunoprecipitated RNAs were further purified with TRIzol LS Reagent (Invitrogen) and treated with 2 U of DNase Turbo (Ambion) in 50 μl reaction buffer for 20 min at 37°C. The immunoprecipitated RNAs were extracted with phenol/chloroform (pH 5.2) and recovered by adding an equal volume of 2-propanol and 1/10 vol of 3 M sodium acetate (pH 5.2) using 1 μl of ethachinmate (Nippon Gene) as a carrier. The precipitated RNAs were dissolved in 50 μl of DEPC-treated water. Samples were denatured for 15 min at 65°C in loading buffer (6.6% formaldehyde, 50% formamide, 20 mM MOPS, 1 mM EDTA, 5 mM sodium acetate, and 0.1 U/μl RNaseOUT) before dot-blot analysis. For the quantitative RIP experiment, the dissolved RNAs were further treated with 2 U of DNase Turbo in 50 μl reaction buffer for 20 min at 37°C, extracted with phenol/chloroform (pH 5.2), and recovered by adding an equal volume of 2-propanol and 1/10 vol of 3 M sodium acetate (pH 5.2). cDNA was synthesized by SuperScript III reverse transcriptase (Invitrogen) with random hexamers. Quantitative

real-time PCR was performed by the Fast SYBR Green Master Mix (Applied Biosystems) and the StepOnePlus Real Time PCR system (Applied Biosystems).

## Chromatin fractionation assays

Cells were harvested by trypsin treatment, washed twice with ice-cold PBS, and resuspended in lysis buffer (50 mM Tris-HCl pH 7.5, 5 mM $MgCl_2$, 100 mM NaCl, 1% Triton-X100, 10% glycerol, and 1 mM sodium fluoride). After 10 min incubation on ice, the lysates were centrifuged at 14,010 $\times$g for 10 min at 4°C to separate supernatant and chromatin-enriched pellet fractions. Whole cell lysates and supernatant and pellet fractions were resuspended in SDS-sample buffer and sonicated using a Bioruptor (Diagenode) to reduce viscosity before SDS-PAGE analysis. Proteins in each fraction were analyzed by western blotting.

## FRAP analysis and Hoechst staining

FRAP was performed as described previously (*Hayashi-Takanaka et al., 2009*) using a confocal microscope (FV-1000; Olympus; operated by the built-in software FV10-ASW ver. 4.2), with a PlanApoN 60$\times$ OSC NA = 1.4 oil-immersion objective lens. iMEFs were grown on a glass-bottom dish (Mat-Tek) set on a heated stage at 37°C (Tokai Hit) in a 5% $CO_2$ atmosphere control system (Tokken). Two images were collected at a 5 s interval (0.8% 488 nm laser transmission; 2.0 µs/pixel; 512 $\times$ 512 pixels; pinhole 800 µm;$\times$8.0 zoom; line averaging 2), a circle (2.55 µm diameter) covering a heterochromatin region was bleached (100% 488 nm laser transmission; 4.0 µs/pixel; 1 iteration), and 25 images were further acquired every 5 s using the original settings. Fluorescence intensities were measured using the ImageJ 1.47v program (http://rsb.info.nih.gov/ij/). The net intensities of the bleached and unbleached heterochromatin regions were obtained by background subtraction in each time frame. The intensity of the bleached heterochromatin was normalized to that of the unbleached heterochromatin, and the intensity relative to the original intensity before bleaching was plotted. The recovery curves from single cells were individually fitted to single exponential kinetics to obtain the half-time ($t_{1/2}$) and mobile fraction (*Kimura and Cook, 2001*), using ImageJ 1.47v. The recovery curve was governed by R = 0.2 + $P$(1-exp$^{-kt}$), where R = relative intensity, $P$ = plateau value, k = association constant, and $t$ = time. The images obtained before bleaching were also used to measure the heterochromatin:euchromatin ratio.

To confirm the enrichment of GFP-Suv39h and its mutants in Hoechst-dense heterochromatin, iMEFs were fixed with 4% paraformaldehyde (Electron Microscopy Sciences) for 10 min and stained with 100 ng/ml Hoechst33342 (Nacalai Tesque) for 1 hr before collecting images with a confocal microscope (FV-1000; Olympus; operated by the built-in software FV10-ASW ver. 4.2), with a PlanApoN 60$\times$ OSC NA = 1.4 oil-immersion objective lens (sequential scanning using 0.1% 405 nm and 4% 488 nm lasers; 2.0 µs/pixel; 800 $\times$ 800 pixels; pinhole 800 µm;$\times$3.0 zoom; line averaging 6).

## Antisense LNA-DNA GapmeR

We obtained locked nucleic acid (LNA)-DNA GapmeRs from Exiqon. An LNA-DNA GapmeR for major satellite RNA was previously reported (*Probst et al., 2010*). The sequence of the Neat1 LNA-DNA GapmeR was ccagTCCACCCGTCtcca (LNA nucleotides in lower case letters). For knockdown experiments with locked nucleic acid (LNA)-GapmeRs, trypsinized Suv39h dn iMEFs ($1 \times 10^6$ ~cells) were suspended in 100 µl MEF II nucleofector solution (Lonza) and then mixed with LNA-DNA GapmeRs (250 nM final concentration)). Transfection was carried out in an electroporation cuvette using the Nucleofector instrument (Lonza). The transfected cells were transferred to fresh medium and incubated at 37°C for 24 hr. Then these cells were changed into fresh medium and incubated at 37°C for 24 hr, followed by harvesting cells for RNA preparation or FRAP analysis.

## Dot-blot analysis

Immunoprecipitated RNA or DNA samples were blotted onto a Hybond N+ nylon membrane (GE Healthcare) by Bio-Dot (Biorad) according to the manufacturer's instructions. The membrane was rinsed with 10X SSC (1.5 M NaCl and 150 mM sodium citrate) and crosslinked with a UV Stratalinker (Stratagene). The membrane was pre-hybridized at 55°C for at least 15 min and hybridized with a probe at 55°C overnight. PCR-amplified major satellite or 18S rRNA probes were labeled and hybridized using the Alkphos Direct Labeling and Detection System (GE Healthcare) according to the

manufacturer's instructions. The membrane was washed three times with a first-wash buffer (2 M Urea, 0.1% SDS, 50 mM NaHPO$_4$ pH 7.0, 150 mM NaCl, 1 mM MgCl$_2$, and 0.1% blocking reagent) at 55°C and twice with a second-wash buffer (50 mM Tris, 100 mM NaCl, and 1 mM MgCl$_2$) at room temperature. Images were captured with a LAS-3000 Mini (Fujifilm).

## Expression and purification of recombinant proteins

A recombinant GST-fusion protein was expressed in *E. coli* BL21 (DE3) cells. Cells were lysed by sonication (Branson, Sonifier 450D), and proteins in the lysates were purified using Glutathione Sepharose (GE Healthcare) according to the manufacturer's instructions. To produce maltose binding protein (MBP)-fused recombinant Suv39h1 (MBP-Suv39h1) proteins, the corresponding Suv39h1 cDNAs were introduced into the pMAL-c2X vector (NEB). Proteins expressed in the *E. coli* strain BL21 (DE3) were purified using Amylose Resin (NEB: E8021S) according to the manufacturer's instructions. The GST- or MBP-fused recombinant proteins were further purified by anion-exchange chromatography (SOURCE 15Q; GE Healthcare). The eluted material was dialyzed against phosphate-buffered saline (PBS) with 10% glycerol, divided into aliquots, and stored at −80°C until use.

## ITC

ITC was performed on a MicroCal VP-ITC calorimeter (GE Healthcare) at 20°C. The proteins were dissolved in phosphate buffer (20 mM KPO$_4$ pH 6.8 and 10 mM NaCl). A typical titration consisted of injecting 10 µl aliquots of the ligand (H3K9me0-3; ARTKQTAR(Lys[Me]0–3)STGGKAPRKQLAY) into the protein sample at 3 min intervals to ensure that the titration peak returned to the baseline. The ITC data were analyzed using the Origin program.

## Peptide pull-down assays

Peptide pull-down assays were performed as described previously (*Ishida et al., 2012*). GST-CD fusion molecules (1 pmol) were incubated with 10 µl of Dynabeads M-280 Streptavidin-conjugated with biotinylated unmodified, K9-monomethylated, K9-dimethylated, or K9-trimethylated H3 peptide (100 pmol) in 150 µl of binding buffer (20 mM Tris-HCl pH 8.0, 150 mM NaCl, 1 mM EDTA, and 0.1% Triton X-100) for 1 hr at 4°C. To examine the relationship between nucleic acid–binding and H3K9me3 recognition, 1 pmol of major satellite RNA or dsDNA probe was added to the binding reaction. The beads were washed twice with binding buffer, and the bound proteins were eluted in SDS sample buffer and analyzed by immunoblotting with an anti-GST antibody.

## Small scale ChIP analysis

A small scale ChIP analysis was performed as described previously (*Brind'Amour et al., 2015*). iMEF cells (1 × 10$^5$ cells) were harvested and re-suspended in nuclear isolation buffer (Sigma). The chromatin was fragmented for 7 min at 37°C, and diluted in ChIP immunoprecipitation buffer (20 mM Tris-HCl pH 8.0, 2 mM EDTA, 150 mM NaCl, 0.1% Triton X-100, 1 × EDTA-free protease inhibitor cocktail and 1 mM PMSF). Lysate mixture was incubated with anti-H3K9me3 (2F3) coupled with 10 µl of mouse IgG Dynabeads for over night at 4°C. The immune complexes were sequentially washed in low salt wash buffer (0.1% SDS, 1% Triton X-100, 2 mM EDTA, 20 mM Tris-HCl pH 8.0, 150 mM NaCl), and high salt wash buffer (0.1% SDS, 1% Triton X-100, 2 mM EDTA, 20 mM Tris-HCl pH 8.0, 500 mM NaCl ). The immune complex was purified with 1.8×volume of Ampure XP DNA purification beads (Agencourt), then analyzed by real-time PCR using primers reported as specific for major satellite DNA (*Matsui et al., 2010*).

## Primer sequence for qRT-PCR

PCR primers used for RT-qPCR were as follows: Major Satellite ChIP F (5'- GACGACTTGAAAAA TGACGAAATC-3'), Major Satellite ChIP R (5'- CATATTCCAGGTCCTTCAGTGTGC-3'), Gapdh ChIP F (5'- ATCCTGTAGGCCAGGTGATG-3'), Gapdh ChIP R (5'- AGGCTCAAGGGCTTTTAAGG-3'), Gapdh RT-PCR F (5'- ATGAATACGGCTACAGCAACAGG-3'), Gapdh RT-PCR R (5'- CTCTTGCTCAG TGTCCTTGCTG-3'), NEAT1-6 F (5'- TGCTTACACGGCTTGTTCAG-3'), NEAT1-6 R (5'- AAC TCCAAGGTCCCTGTCCT-3').

## Cell proliferation assay

*Suv39h* dn iMEFs were seeded onto a 10 cm dish the day before the experiment. Cells were labeled with 10 µM CellTrace Far Red (Thermo Fisher) on the dish for 20 min at 37°C, then washed twice in medium and returned to the incubator for 10 min at 37°C. Then, these labeled *Suv39h* dn iMEFs were infected with a retrovirus containing pMCs-FLAG-Suv39h1$^{WT/mut}$-IRES-GFP, and incubated for 3 days at 37°C. Analysis was performed on 10,000 cells/ GFP$^{dull+}$ sample using FACS.

## Immunopurification of FLAG-tagged Suv39h1 from HEK293T

48 hr after transfection, 293T cells expressing FLAG-tagged Suv39h1$^{wt/mut}$ were lysed in RIPA buffer (50 mM Tris-HCl pH 7.5, 420 mM NaCl, 1 mM EDTA, 1% NP40, and 0.25% sodium deoxycholate) containing 1 mM PMSF, and were subjected to sonication on ice. FLAG-tagged Suv39h1$^{wt/mut}$ proteins were purified with anti-FLAG M2 affinity gel.

## *In vitro* histone methyltransferase assay

An *in vitro* histone methyltransferase assay was performed as described previously (*Shimazu et al., 2014*). FLAG-tagged wild-type or mutant Suv39h1 was incubated in 1x Collin's buffer (25 mM Tris-HCl pH 8.5, 5 mM DTT) with Histone H3 (0.5 µg) and 14C-labeled SAM (0.01 µCi, Perkin Elmer) for 15 min at 25°C. The reaction was stopped by adding Laemmli SDS sample buffer. Proteins were resolved on a 15% acrylamide SDS-PAGE gel, and the dry gel was exposed to an imaging plate (Fujifilm) for 1 day. Autoradiography was detected using a BAS-5000 Image Analyzer (Fujifilm).

## Acknowledgements

We thank T Jenuwein for providing the *Suv39h* dn iMEFs. We also thank K Kotoshiba and M Kato for their technical help, H Tagami for his helpful comments on the manuscript, T Umehara and M Kikuchi for their advice on the Suv39h1-CD mutational studies and EC. Bradshaw for English editing. Finally, we thank all of the Shinkai lab members for helpful discussions and the Support Unit for Bio-Material Analysis, RIKEN BSI Research Resources Center, with special thanks to Mr. K Ohtawa for FACS analysis and cell sorting. The authors have no financial interests related to this work. This research was supported by KAKENHI (23114005 and 26291072) for JN, a RIKEN internal research fund for YS, KAKENHI (25116005) for HK and the NMR platform (07022019) and the Platform for Drug Discovery, Informatics, and Structural Life Science (12743018) for YN. AS was supported by RIKEN Subsidies to pay for a research support person for research staff with family responsibilities.

## Additional information

### Funding

| Funder | Grant reference number | Author |
| --- | --- | --- |
| Drug Discovery, Informatics, and Structural Life Science | 12743018 | Yoshifumi Nishimura |
| Ministry of Education, Culture, Sports, Science and Technology | 07022019 | Yoshifumi Nishimura |
| Ministry of Education, Culture, Sports, Science and Technology | 25116005 | Hiroshi Kimura |
| Ministry of Education, Culture, Sports, Science and Technology | 23114005 | Jun-ichi Nakayama |
| Ministry of Education, Culture, Sports, Science and Technology | 26291072 | Jun-ichi Nakayama |
| RIKEN | | Yoichi Shinkai |

The funders had no role in study design, data collection and interpretation, or the decision to submit the work for publication.

## Author contributions
AS, Supervision, Funding acquisition, Investigation, Writing—original draft, Project administration; TK, Investigation, Writing—original draft; HS, DM, MI-Y, Investigation; YN, Supervision, Funding acquisition; HK, J-iN, YS, Supervision, Funding acquisition, Investigation, Writing—original draft

## Author ORCIDs
Atsuko Shirai, http://orcid.org/0000-0003-1344-4746
Hiroshi Kimura, http://orcid.org/0000-0003-0854-083X
Jun-ichi Nakayama, http://orcid.org/0000-0002-5597-8239
Yoichi Shinkai, http://orcid.org/0000-0002-6051-2484

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
