## [Decision Letter]

Thank you for submitting your article "Impact of nucleic acid and methylated H3K9 binding activities of Suv39h1 in its heterochromatin assembly" for consideration by *eLife*. Your article has been favorably evaluated by three reviewers, one of whom, Jessica Tyler (Senior Editor), is a member of our Board of Reviewing Editors. The reviewers have opted to remain anonymous.

The reviewers have discussed the reviews with one another and the Reviewing Editor has drafted this decision to help you prepare a revised submission.

Summary:

This study explores the possible function of an RNA binding activity in the chromo domain of the mouse Suv39h1 in its heterochromatin localization. The authors demonstrate that the chromo domain of Suv39h1 binds to single stranded RNA (ssRNA), has higher affinity for ssRNA than dsDNA, generate mutations in the chromo domain that abolish RNA binding without affecting the association of the chromo domain with H3K9me3 peptides, show that Suv39h1 associates with major satellite RNA in vivo, and that RNA binding appears to affect the stable association of Suv39h1 with heterochromatic foci. The experiments in the paper are well executed and support the main conclusion regarding the effect of RNA binding on the stable association of at least Suv39h1-EGFP (which is somewhat overexpressed) with heterochromatin. Non-specific affinity for RNA (or DNA) may potentially play a role in heterochromatin stability. However, the functional significance of the observed non-specific RNA binding (effects on silencing) is not established and additional work is required to rule out alternative explanations prior to publication.

Essential revisions:

1) The authors present some data on the effect of the RNA binding mutations on the levels of H3K9me3 and HP1 β at heterochromatic foci. Here the effect is weak, 2-fold or less reduction in intensity in the mutants (Figure 3). They should also test whether RNA binding is required for silencing of major satellite repeats and/or other transposons by introducing wild type versus mutant Suv39h1 into Suv39h1/h2 DKO MEFs. For H3K9me3 levels, rather than the crude assay involving intensity of staining at heterochromatic foci, the authors should perform ChIP.

2) Figure 1. Considering MSR's are produced in both sense and antisense orientations (Bulut-Karslioglu et al. 2012), binding affinity to double stranded (ds) MSR RNAs also should be tested.

3) Subsection “Suv39h1-CD's nucleic acid binding and methylated H3K9 recognition/binding are both crucial for Suv39h1-mediated pericentric heterochromatin assembly”/Figure 3: Is there any growth defect in Suv39 dn with mutant Suv39h1? Figure 3 suggest that Suv39h1 mutants display a slower kinetics in reestablishing H3K9me at day 3. One possibility, which needs to be ruled out, is that the overall process (which is cell cycle regulated – high MSR expression in G1/S phase) proceeds at the same pace between wild type and mutant cell lines, but because mutant cells have slower doubling times, the mutant cells have not undergone the same number of doubling times thus appear to show fewer H3K9me foci. Normalization by doubling time will rule out this possibility and provide strong evidence that the mutants' reestablishment kinetics is slower than wt.

4) Figure 2—figure supplement 1: Why does addition of unmodified H3 result in loss of Chp1 binding to ssRNAs? This also seems to occur with Suv39h1-CD.

5) Subsection “Suv39h1-CD’s nucleic acid binding is required for its interaction with major satellite RNAs in vivo”/Figure 4: The authors develop great tools with which they can distinguish between CD's ability to bind to H3K9me versus ssRNA. The data in Figure 4 show that Suv39h1 interaction with MSR RNAs is reduced in 4A mutant, but it is unclear whether RNA binding and H3K9me binding make additive contribution to this interaction. For example, it is also possible that Suv39h1 localizes to MSRs via H3K9me3 recognition first, followed by binding to MSR RNAs. The following experiments will help address this issue:

A) What happens to RIP of Y67A and Y67A-4A? A prediction of the proposed model is that combining RNA- and H3K9me binding mutations in the same protein should reduce the ability of Suv39h1 to interact with MSR RNAs more than either of the single mutants. For better quantification, it is suggested to consider performing qRT-PCR reactions and normalize all RIP signals to input RNA.

B) These RIP experiments could be performed from nuclear and cytoplasmic fractions to ask whether Suv39h1 interaction with RNA occurs in nucleus and requires H3K9me3 (in case of Y67A).

6) Subsection “Full-length Suv39h1 exhibits diminished nucleic-acid binding activity *in vitro*”/Figure 1—figure supplement 1: RNA binding affinity of full-length protein may be similar to Suv39h1 CD. The authors are encouraged to re-examine their*in vitro* binding assay results. Figure 1—figure supplement 1 and L reveal the formation of a large band, which fails to enter the gel. The accumulation of this large band correlates with the disappearance of ssRNA substrate (lanes, 6 and 7 from the left) which may suggest a similar binding kinetics for the full-length protein and CD alone. This putative larger product may enter the gel by running them longer using lower voltage. This may reveal that the full-length and CD regions alone have similar binding kinetics to ssRNA substrates.

7) Figure 5—figure supplement 1. Why does mut1, which shows no significant defect in binding to MSR ssRNA *in vitro* (Figure 1), display rapid mobility comparable to mut3 (and other ssRNA-binding) mutants? This appears inconsistent with the hypothesis that loss of ssRNA binding results in lower residency of Suv39h1 with heterochromatin.

8) Figure 5 shows no clear reduction in Suv39h1 interaction with chromatin until both the H3K9me3 and RNA binding interactions are disrupted. However, the figure legend states that each interaction is crucial for targeting Suv39h1. Also, the text says that signal was increased in the euchromatin for the single mutants. This is not apparent from the data shown in Figure 5. Even with both regions mutated, significant H3K9me3 is retained at the heterochromatin. These overinterpretations need to be toned down.

9) Showing that RNaseH / A treatment has a similar effect to the Suv39h1 nucleic acid binding mutation in the FRAP analyses would make the results more convincing that the RNA is a key contributor to Suv39h1 stability on chromatin.

---

## [Author Response]

Essential revisions:

1) The authors present some data on the effect of the RNA binding mutations on the levels of H3K9me3 and HP1 β at heterochromatic foci. Here the effect is weak, 2-fold or less reduction in intensity in the mutants (Figure 3). They should also test whether RNA binding is required for silencing of major satellite repeats and/or other transposons by introducing wild type versus mutant Suv39h1 into Suv39h1/h2 DKO MEFs.

Thank you for this comment. Since Suv39h is mostly dispensable for silencing of retrotransposons in iMEFs (Bulut-Karslioglu et al., Mol Cell, 2014), we focused on major satellite repeats (MSRs). We compared MSR expression levels in wild-type (WT) and *Suv39h* dn iMEFs that were infected with an empty or expression vector for Suv39h1 (WT and mutants) on day 14. As reported previously (Bulut-Karslioglu et al., Nat Struct Mol Biol. 2012), MSRs were derepressed in *Suv39h* dn iMEFs (~10 times higher than that of WT iMEFs) and introduction of WT Suv39h1 suppressed the up-regulation of MSR expression. Mutant Suv39h1 proteins (i.e., Y67A, 4A, and Y67A-4A) also decreased MSR expression, but their effects were milder than that of WT, which is consistent with their phenotypes of pericentric accumulation of H3K9me3 and HP1β. Furthermore, among these mutants, Suv39h1-Y67A-4A exhibited the weakest effect, which is also consistent with the H3K9me3/HP1β IF staining and GFP-Suv39h1 FRAP results.

We included this data in new Figure 3—figure supplement 1 and described this result in the text (subsection “Both Suv39h1-CD’s nucleic acid binding and methylated H3K9 recognition/binding contribute to Suv39h1’s retention on pericentric heterochromatin”, fourth paragraph).

For H3K9me3 levels, rather than the crude assay involving intensity of staining at heterochromatic foci, the authors should perform ChIP.

We performed H3K9me3 ChIP-qPCR analysis on MSRs in *Suv39h* dn iMEFs infected with an empty or expression vector for FLAG-tagged Suv39h1 (WT and mutants). As shown in new Figure 3—figure supplement 13K9me3 level on MSRs was severely diminished in *Suv39h* dn iMEFs and rescued by FLAG-tagged Suv39h1-WT expression. The expression of Suv39h1 mutant Y67A, 4A and Y67A-4A also recovered H3K9me3 levels on day 6 after transduction up the similar levels to that of Suv39h1-WT. Among them, Suv39h1-Y67A-4A showed the weakest recovery, which is consistent with the observations of MSR RT-qPCR, H3K9me3/HP1β IF staining, and GFP-Suv39h1 FRAP analyses.

We included this data in new Figure 3—figure supplement 1 and described this result in the text (subsection “Both Suv39h1-CD’s nucleic acid binding and methylated H3K9 recognition/binding contribute to Suv39h1’s retention on pericentric heterochromatin”, second paragraph).

2) Figure 1. Considering MSR's are produced in both sense and antisense orientations (Bulut-Karslioglu et al. 2012), binding affinity to double stranded (ds) MSR RNAs also should be tested.

We have prepared double stranded MSR RNA and performed EMSA using GST-Suv39h1-CD. As shown in new Figure 1—figure supplement 1, Suv39h1-CD also bound to dsRNA with a comparable binding affinity to ssRNA (compare Figure 1 and Figure 1—figure supplement 1). This result supports an idea that MSR dsRNA can also be a target of Suv39h1, although it is unlikely that dsRNA is a preferential target over ssRNA. We have described this in our revised manuscript (subsection “Suv39h1-CD can bind nucleic acids”, last paragraph).

3) Subsection “Suv39h1-CD's nucleic acid binding and methylated H3K9 recognition/binding are both crucial for Suv39h1-mediated pericentric heterochromatin assembly”/Figure 3: Is there any growth defect in Suv39 dn with mutant Suv39h1? Figure 3 suggest that Suv39h1 mutants display a slower kinetics in reestablishing H3K9me at day 3. One possibility, which needs to be ruled out, is that the overall process (which is cell cycle regulated – high MSR expression in G1/S phase) proceeds at the same pace between wild type and mutant cell lines, but because mutant cells have slower doubling times, the mutant cells have not undergone the same number of doubling times thus appear to show fewer H3K9me foci. Normalization by doubling time will rule out this possibility and provide strong evidence that the mutants' reestablishment kinetics is slower than wt.

Again, thank you for such a valuable comment. for this purpose, we performed celltrace™ (thermo fisher scientific) staining analysis, which is well established method for *in vitro* and *in vivo* labeling of cells to trace multiple generations using dye dilution by flow cytometry.

https://www.thermofisher.com/order/catalog/product/c34564

*Suv39h* dn imefs were labeled with celltrace™ far red, infected with retroviruses for gfp-suv39h1 wt and mutant expression, grown for 3 days, and subjected for facs analysis. far red signal intensities were measured in the gated fraction based on gfp expression level, as demonstrated for the rescue experiments shown in Figure 3. there was little or no difference in far red signal distribution among the *suv39h* dn imefs transduced with different constructs, indicating that the numbers of cell division during 3 days after the virus infection were not much different among them. it is therefore likely that the mutants’ slower heterochromatin re-establishment is not due to the defect or delay in cell cycle progression.

We include this data in new Figure 3—figure supplement 2 and described this result in the text (subsection “both suv39h1-cd’s nucleic acid binding and methylated h3k9 recognition/binding contribute to suv39h1’s retention on pericentric heterochromatin”, second paragraph).

4) Figure 2—figure supplement 1: Why does addition of unmodified H3 result in loss of Chp1 binding to ssRNAs? This also seems to occur with Suv39h1-CD.

Since Chp1-CD does not bind to unmodified H3 peptide (Ishida et al. Mol Cell, 2012, Figure 2), loss of Chp1-CD binding to ssRNAs in the presence of unmodified H3 was likely due to an electrostatic interaction between unmodified H3 and the RNA probe that inhibited the ability of Chp1-CD to bind the probe. Such an effect was described in the previous study (Ishida et al. Mol Cell, 2012, Figure 3). While Suv39h1-CD weakly bound to unmodified H3 peptide (new Figure 2), this negative effect was also likely due to an electrostatic interaction between unmodified H3 and the RNA probe, and the negative effect of H3 peptides appears to be alleviated in the case of K9me3 peptide since Suv39h1-CD binds to the H3 peptide. We now carefully explain the negative effect in our revised manuscript (subsection “Suv39h1-CD’s RNA binding and methylated H3K9 recognition/binding are independent of each other”, second paragraph).

*5) Subsection “Suv39h1-CD’s nucleic acid binding is required for its interaction with major satellite RNAs in vivo”/Figure 4: The authors develop great tools with which they can distinguish between CD's ability to bind to H3K9me versus ssRNA. The data in Figure 4 show that Suv39h1 interaction with MSR RNAs is reduced in 4A mutant, but it is unclear whether RNA binding and H3K9me binding make additive contribution to this interaction. For example, it is also possible that Suv39h1 localizes to MSRs via H3K9me3 recognition first, followed by binding to MSR RNAs. The following experiments will help address this issue:*

A) What happens to RIP of Y67A and Y67A-4A? A prediction of the proposed model is that combining RNA- and H3K9me binding mutations in the same protein should reduce the ability of Suv39h1 to interact with MSR RNAs more than either of the single mutants. For better quantification, it is suggested to consider performing qRT-PCR reactions and normalize all RIP signals to input RNA.

As suggested by reviewers, we have performed RIP experiments using cells expressing wild-type or mutant Suv39h1 and examined interaction with MSR RNAs by RT-qPCR. We have confirmed that Suv39h1-WT stably bound MSR ssRNAs and that this interaction was impaired for Suv39h1-4A (Figure 4—figure supplement 1). We found that this interaction was also lost for Suv39h1-Y67A mutant, suggesting that H3K9me3 recognition by Suv39h1-CD also required for the Suv39h1’s stable interaction with MSR RNAs. Intriguingly, Suv39h1-Y67A-4A interacted partially with MSR RNAs. This result does not seemingly fit with our model that Suv39h1-CD’s RNA- and H3K9me3-binding activities cooperatively function in Suv39h1’s chromatin targeting. However, we also found that Suv39h1-Y67A-4A loosely bound chromatin (see below). Considering this observation, Suv39h1-Y67A-4A may interact cellular RNAs without stable binding to chromatin. We have included these new results in Figure 4 and discussed possible interpretation in the revised manuscript (subsection “Suv39h1-CD’s nucleic acid binding is required for its interaction with major satellite RNAs *in vivo*”, last paragraph).

B) These RIP experiments could be performed from nuclear and cytoplasmic fractions to ask whether Suv39h1 interaction with RNA occurs in nucleus and requires H3K9me3 (in case of Y67A).

In the RIP experiments, cells were fixed with 1% formaldehyde before harvest and we have no experience to fractionate cells after fixation. To address whether Suv39h1 interaction with RNA occurs in nucleus, we conducted chromatin fractionation assay without fixation and examined Suv39h1’s chromatin association. We found that Suv39h1 was predominantly detected in the chromatin-enriched precipitated fractions and only negligibly in the soluble fractions containing cytoplasmic proteins (new Figure 4). While Suv39h1 mutants (4A and Y67A) failed to bind MSR RNAs in the RIP experiments, most of these mutants were also detected in the chromatin-enriched precipitated fractions. These results support an idea that Suv39h1 interaction with RNA occurs in nucleus and also suggest that mutant Suv39h1 somehow retained chromatin association without stable binding with MSR RNAs. Interestingly, a relatively higher level of Suv39h1-Y67A-4A was detected in the soluble fractions. Suv39h1-Y67A-4A’s weakened interaction with MSR RNAs and trimethylated H3K9 might be linked with its impaired localization. We have described these results in the revised manuscript (subsection “Suv39h1-CD’s nucleic acid binding is required for its interaction with major satellite RNAs *in vivo*”, last paragraph).

6) Subsection “Full-length Suv39h1 exhibits diminished nucleic-acid binding activity in vitro”/Figure 1—figure supplement 1: RNA binding affinity of full-length protein may be similar to Suv39h1 CD. The authors are encouraged to re-examine their in vitro binding assay results. Figure 1—figure supplement 1 and L reveal the formation of a large band, which fails to enter the gel. The accumulation of this large band correlates with the disappearance of ssRNA substrate (lanes, 6 and 7 from the left) which may suggest a similar binding kinetics for the full-length protein and CD alone. This putative larger product may enter the gel by running them longer using lower voltage. This may reveal that the full-length and CD regions alone have similar binding kinetics to ssRNA substrates.

We thank the reviewer for raising this point. One of the obstacles that hampered our EMSAs using full-length Suv39h1 proteins was their low solubility: most of recombinant Suv39h1 proteins went into insoluble fractions, and we had to use a harsh condition to obtain sufficient amount of full-length Suv39h1. We speculate that this procedure might compromise the function of full length Suv39h1 and affect the EMSA experiments. We noticed, however, that the solubility of full-length Suv39h1 was dramatically improved when it was expressed with a maltose binding protein (MBP). Using this system, we have prepared MBP alone, MBP-fused Suv39h1-CD, and full-length Suv39h1 WT and mutant (4A) (new Figure 1—figure supplement 1) and conducted EMSAs. We found that MBP-fused full-length Suv39h1 bound ssRNA with a higher affinity compared with MBP-Suv39h1-CD, and 4A mutation reduced the binding affinity of full-length Suv39h1 (Figure 1—figure supplement 1). These results suggest that full-length Suv39h1 can bind to ssRNA and also imply that Suv39h1’s domains other than CD also contribute to its ssRNA binding. Considering partial effects of Suv39h1 mutations (Figure 3 and Figure 5), it is possible that ssRNA-binding activity associated with Suv39h1’s domains other than CD participates in its chromatin targeting. While this would be important for considering the Suv39h1’s chromatin targeting mechanism, we feel that this is currently beyond the scope of the current manuscript. However, we have added a comment on this issue in our revised manuscript (subsection “Full-length Suv39h1 exhibits diminished nucleic-acid binding activity*in vitro*”).

7) Figure 5—figure supplement 1. Why does mut1, which shows no significant defect in binding to MSR ssRNA in vitro (Figure 1), display rapid mobility comparable to mut3 (and other ssRNA-binding) mutants? This appears inconsistent with the hypothesis that loss of ssRNA binding results in lower residency of Suv39h1 with heterochromatin.

We thank the reviewer for raising this important point. Yes, the mut1’s behavior in the FRAP experiment (Figure 5—figure supplement 1) seems to be inconsistent with its RNA-binding ability in the EMSA(new Figure 2). However, although mut1 could induce band shift, the titration EMSAs (new Figure 2) clearly shows the difference between WT and mut1 (Figure 2, compare red and green lines). Therefore, as stated in the text “In EMSAs, a Suv39h1-CD mutant (mut1) with N-terminal distal amino acid substitutions bound RNA more weakly than wild-type Suv39h1-CD”, we consider that mut1 is weakened for RNA binding, and this weakened RNA-binding ability led to the increased Suv39h1’s mobility in the FRAP analysis. However, we could not fully explain why mut1 and other RNA binding defective Suv39h1 mutants showed similar FRAP profiles despite their differences in RNA binding activity. Impacts of mut1 on full-length Suv39h1’s RNA binding might be enhanced in *in vivo* context. Alternatively, impacts on mut1 on Suv39h1-CD’s RNA-binding might be compensated by *in vitro* condition such as GST’s dimerization property. In either case, we believe our *in vitro* RNA binding data still maintains our hypothesis.

8) Figure 5 shows no clear reduction in Suv39h1 interaction with chromatin until both the H3K9me3 and RNA binding interactions are disrupted. However, the figure legend states that each interaction is crucial for targeting Suv39h1. Also, the text says that signal was increased in the euchromatin for the single mutants. This is not apparent from the data shown in Figure 5. Even with both regions mutated, significant H3K9me3 is retained at the heterochromatin. These overinterpretations need to be toned down.

Based on the reviewer’s suggestion, we revised Figure 5 title as follows:

Original: Figure 5. Suv39h1-CD’s nucleic acid binding and H3K9me binding are both crucial for targeting Suv39h1

Revised: Figure 5. Both Suv39h1-CD’s nucleic acid binding and H3K9me binding contribute to Suv39h1’s retention on heterochromatin

Regarding the statement in the original text, we rephrased the text to be more precise.

“GFP-Suv39h1-4A, -Y67A, and -Y67A-4A also formed foci on Hoechst-dense regions (Figure 5). However, semi-quantitative analysis to measure their heterochromatin enrichment (represented as the intensity ratio of euchromatin over heterochromatin; Figure 5) revealed that GFP-Suv39h1-Y67A-4A fluorescence was significantly increased in euchromatin compared to the wt (Figure 5), suggesting that Suv39h1-Y67A-4A’s affinity to heterochromatic regions was decreased. GFP-Suv39h1-Y67A was also slightly less enriched on heterochromatin, and Y67A and 4A appeared to have an additive effect.”

9) Showing that RNaseH / A treatment has a similar effect to the Suv39h1 nucleic acid binding mutation in the FRAP analyses would make the results more convincing that the RNA is a key contributor to Suv39h1 stability on chromatin.

We have tried RNaseH/A treatments by permeabilized cells, but it has been difficult to interpret the results. Therefore, we performed knock down experiments to reduce major satellite repeat (MSR) transcripts using locked nucleic acid (LNA)-DNA gapmers (Probst et al., Dev Cell, 2010). Introduction of MSR-specific LNA-DNA gapmers significantly decreased MSR transcripts in *Suv39h* dn iMEFs and enhanced the mobility of GFP-Suv39h1 on pericentric heterochromatin. This data further supports that MSR RNA contributes to Suv39h1 stability on pericentric regions.

We added this data as new Figure 6 and described the result in the new section “Major satellite RNAs contribute to retention of Suv39h1 on heterochromatin”.

In this RNA KD experiment, we also included nuclear non-coding RNA, Neat1 specific LNA-DNA gapmers as a negative control. Neat1 specific LNA-DNA gapmers decreased *Neat1* but not MSRs transcripts, but did not significantly enhance dynamics of GFP-Suv39h1 on heterochromatin, indicating the specificity of effects of MSRs specific LNA-DNA gapmer. However, due to the time limit, we could only complete this experiment including the cells treated with *Neat1* specific LNA-DNA gapmers once. Therefore, we provide this data (Figure 7) only for the reviewing purpose.

Author response image 1.Major satellite RNAs contribute to Suv39h1’ s retention on heterochromatin.(**A**) qRT-PCR for major satellite RNAs or nuclear non-cording RNA NEAT1 transcripts in Suv39h dn iMEFs expressing GFP-suv39h1 transfected without or with LNA-DNA GapmeRs directed NEAT1 transcript or set of two LNA-DNA GapmeRs directed against the Forward and Reverse major satellite transcripts. (**B**) FRAP analysis of GFP-Suv39h1 in the described cells (**A**). The means of the relative intensity in the bleached area are indicated with the SD (n ≥ 13).**DOI:**
http://dx.doi.org/10.7554/eLife.25317.017